# Modern anthropogenic drought in Central Brazil unprecedented during last 700 years

Nicolas Misailidis Stríkis [1,2] ✉, Plácido Fabrício Silva Melo Buarque[1,2,3,4], Francisco William Cruz[1], Juan Pablo Bernal [5], Mathias Vuille [6], Ernesto Tejedor[7], Matheus Simões Santos[2], Marília Harumi Shimizu [8], Angela Ampuero [1], Wenjing Du[9], Gilvan Sampaio[8], Hamilton dos Reis Sales[10], José Leandro Campos [1], Mary Toshie Kayano [8], James Apaèstegui [11,12], Roger R. Fu[13], Hai Cheng[9], R. Lawrence Edwards[14], Victor Chavez Mayta[15], Danielle da Silva Francischini[16], Marco Aurélio Zezzi Arruda[16] & Valdir Felipe Novello [17]

A better understanding of the relative roles of internal climate variability and external contributions, from both natural (solar, volcanic) and anthropogenic greenhouse gas forcing, is important to better project future hydrologic changes. Changes in the evaporative demand play a central role in this context, particularly in tropical areas characterized by high precipitation seasonality, such as the tropical savannah and semi-desertic biomes. Here we present a set of geochemical proxies in speleothems from a well-ventilated cave located in central-eastern Brazil which shows that the evaporative demand is no longer being met by precipitation, leading to a hydrological deficit. A marked change in the hydrologic balance in central-eastern Brazil, caused by a severe warming trend, can be identified, starting in the 1970s. Our findings show that the current aridity has no analog over the last 720 years. A detection and attribution study indicates that this trend is mostly driven by anthropogenic forcing and cannot be explained by natural factors alone. These results reinforce the premise of a severe long-term drought in the subtropics of eastern South America that will likely be further exacerbated in the future given its apparent connection to increased greenhouse gas emissions.

Since the 1970s, observational records and models point to a widespread drying trend affecting many land areas across the globe[1–5]. Drought events impact societies and their countries' economies in multiple ways, leading to agricultural losses, reduced energy production from hydropower, and affecting public water supply systems[6,7]. Intensive agriculture is highly dependent on water availability and, usually, is the first and most heavily affected sector impacted by drought, with important implications for food security. Accurate projections of future drought risk are crucial for policy formulation and implementation of adaptation and mitigation strategies and for effective disaster and risk reduction policy[6]. Regions within southern

Amazonia and central–eastern Brazil are now characterized by a long-term hydrologic drought, which appears to be highly correlated with changes in surface temperature[5,8–10]. Central Brazil plays an important role in the country´s economy, due to its intensive agriculture, but it is also home to important ecosystems. The increased aridity observed in the region during the last decades has raised questions regarding the role of human-induced warming in driving changes in the hydrologic balance and how drought risk might be exacerbated by anthropogenic greenhouse gas emissions in the coming decades[2,3,9,11].

In general, a drought can be defined as a period with a hydrologic deficit, when the water evapotranspiration exceeds the input from

effective precipitation[4,12]. This type of drought is also commonly known as "hydrologic drought" and its effects are most apparent in reduced streamflow. Hence, in its simplest form, a hydrologic drought can be represented by a simple hydrologic balance model based on precipitation (P) and potential evapotranspiration (PET) according to the following formulation: Hydrologic Balance = P-PET[1,13,14]. Future projections of drought risk, however, are a major challenge for the climate community. At the local scale, temperature plays an important role in the hydrologic balance, with warmer temperature leading to an increase in evapotranspiration and a decrease in soil moisture. On a larger scale, changes in sea surface temperature also influence land precipitation, driving hydroclimate variability at interannual to multi-decadal timescales[15–17].

Reconstructing past climatic conditions in the region, prior to the period covered by instrumental records, is vital to document the full amplitude of natural climate variability and to quantify the contributions from external and internal forcings to regional hydrologic variability[11]. Speleothems can be very useful in this regard, providing high-resolution hydroclimate reconstructions with accurate age control spanning decades to hundreds of thousands of years. Drought events can affect the chemistry and the isotopic composition of speleothems in well-ventilated cave rooms. In such rooms, isotopic and geochemical exchange between drip water and cave atmosphere allow reconstructing past variations of the atmospheric conditions associated with local potential evapotranspiration (PET) and precipitation variability[18–20]. However, few paleoclimate studies have investigated the potential of speleothems from well-ventilated cave rooms to record past variations of the local environmental conditions[21]. Speleothem $\delta^{18}O$ records from tropical regions have been used to reconstruct past monsoon activity and rainfall amount[22] by relying on speleothems collected in isolated chambers deep inside caves, to avoid any atmospheric influence from outside the cave. However, in ventilated cave environments, the loss of water by means of evaporation during the dripping process significantly affects the oxygen isotopic composition ($\delta^{18}O_{calcite}$)[23]. Hence, both a reduction in the amount of rainfall and an increase in evaporation will result in higher $\delta^{18}O_{calcite}$ values of the speleothem. Similarly, dry conditions related to reduced cave dripping and atmospheric relative humidity affect speleothem carbon isotopes in the same direction, resulting in higher $\delta^{13}C_{calcite}$[23,24]. Reduced precipitation and decreasing humidity of the cave atmosphere also affect the trace element composition of the speleothems by inducing the so-called "Prior Calcite Precipitation" (PCP) effect. PCP simply refers to the fact that carbonates will precipitate before the drip water reaches the stalagmites[25]. The PCP increases under dry conditions, and the process leaves the remaining solution enriched in both $^{13}C$ and in incompatible trace elements[24–28]. As a result of the PCP, the $\delta^{13}C$ values and trace element ratios, such as Mg/Ca, Sr/Ca, and Ba/Ca, tend to increase in the forming stalagmite during dry periods. When combined, oxygen isotope, carbon and trace elements from near-cave-entrance speleothems provide a powerful tool that allows reconstructing variations of the hydrologic balance (precipitation-evapotranspiration) back in time, beyond the instrumental period, thereby yielding unique, annually resolved records of tropical hydrologic balance (e.g., Bernal et al.[29]).

To put currently observed changes in the hydrologic regime in a long-term context, here we present a multi-proxy reconstruction of annually resolved oxygen and carbon isotopes ($\delta^{18}O$ and $\delta^{13}C$) and trace elements concentrations in speleothems from a cave located in central–eastern Brazil. Our results provide robust evidence that the current drying trend has no precedent in the last seven hundred years. By applying a detection and attribution approach, we demonstrate that anthropogenic forcing is the main driver of the current hydrologic regime, responsible for the observed hydrological deficit in the region through enhanced evapotranspiration combined with a negative precipitation trend.

## Results and discussion
### Study location and methodological approach
The study site is located in central–eastern Brazil, in a tropical Savannah-like biome in the northern part of Minas Gerais State (Fig. 1). The climate in this region is characterized by a strong precipitation seasonality, with an annual mean rainfall of near 950 mm. The wet season occurs during the South American Summer Monsoon (SASM) season, between October and April, and is followed by a pronounced 5-month dry season from May to September. The dry season accounts for only 3% of the total annual precipitation and can last as long as 6–7 months in some years based on historical records from local stations available in the Instituto Nacional de Meteorologia (INMET) in operation since 1915. Due to its prolonged dry season, central Brazil is highly dependent on a reliable wet season and accordingly very vulnerable to variations in precipitation.

To characterize drought variability in central–eastern Brazil we used the instrumental records of precipitation and temperature from local meteorological stations to calculate the P-PET index. The meteorological stations were compiled over an area of nearly $4° \times 4°$, centered over the cave site (Supplementary Table S1). Local precipitation was calculated as the mean of monthly rainfall from all meteorological stations listed in Supplementary Table S1, obtained from the INMET and the Agência Nacional de Águas (ANA). The station-based precipitation data were compared with gridded precipitation data at 1° resolution from the Global Precipitation Climatology Centre (GPCC), version 2018 (Supplementary Fig. S1). The potential evapotranspiration (PET) was calculated based on the FAO-56 Penman–Monteith equation using regional gridded data from ground-based stations provided by Brazilian Daily Weather Gridded Data (BR-DWGD)[30] (Fig. 1). In addition, we used measurements of potential evaporation from Piche evaporimeters from local meteorological stations to assess the local atmospheric evaporative demand. These records were used for comparison with speleothem proxies that are sensitive to local atmospheric evaporative conditions.

Since streamflow integrates the net hydrologic balance over a wider catchment area, we compared the calculated P-PET index with the z-scored streamflow data of local rivers obtained from the ANA (Fig. 1 and Supplementary Table S1). This approach permits us to assess how variations in precipitation and evapotranspiration demand relate to streamflow on a regional scale (Fig. 1). The average streamflow was calculated based on the mean monthly values of each fluviometric station. To determine the onset of the drying trend, we applied the change point detection algorithm[31] to the streamflow time series (see "Methods") (Supplementary Fig. S2). Next, we calculated the streamflow z-score of each station prior to averaging the monthly data. Finally, the reconstructed hydroclimate variability obtained from our speleothem record was compared with a gridded surface temperature reconstruction from the Paleo Hydrodynamics Data Assimilation (PHYDA) product[32]. PHYDA combines annually resolved proxy data with the physical constraints from the Community Earth System Model–Last Millennium Ensemble (CESM-LME) during the past 2000 years. Here we compared the PHYDA regional surface temperature (12–16° S; 41–47° W) with local observational data from the INMET meteorological stations. The two data sets show a strong coherence, and are highly correlated ($r = 0.73$, $P$ value < 0.05) (Supplementary Fig. S3).

To reconstruct the local evapotranspiration demand we combine two speleothems from near the entrance of the Onça Cave (Fig. 1). This cave is located in the karst valley formed by the Peruaçu River (15°05'19.86"S– 44°14'41.39"W), one of the few permanent tributaries of the São Francisco River, which is the fifth largest river of Brazil, extending along 2860 km north-south along central–eastern and Northeast Brazil.

The Onça Cave lies at the bottom of a karst canyon, below ~100 m of limestone, which serves as a water reservoir for the cave dripping

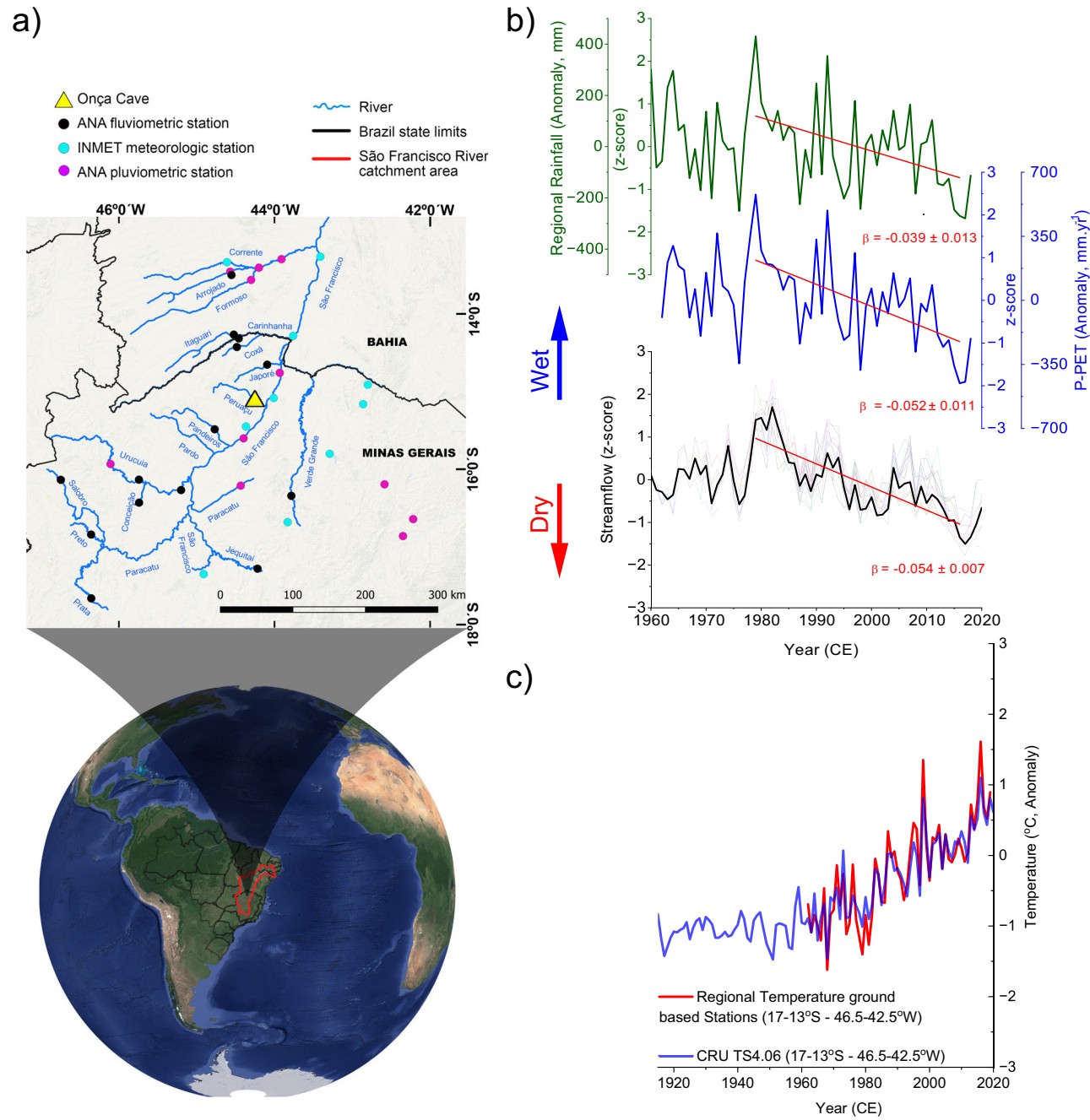

**Fig. 1 | Study site location and the local hydrologic balance. a** Location of the study site and the meteorological and fluviometric stations from the Instituto Nacional de Meteorologia (INMET) and the Agência Nacional de Águas (ANA), respectively, used to calculate streamflow, regional precipitation and hydrologic balance (P-PET index). Satellite image of the Global map: Map data ©2015 Google. Also, see Supplementary Table S1 for station locations. **b** Comparison between time series of (top): z-scored regional precipitation (mean values from INMET and ANA meteorological station, green line), (middle): hydrologic balance calculated as precipitation−potential evapotranspiration (P-PET) (blue line) and (bottom): z-scored streamflow of individual rivers (thin colored lines) and their arithmetic median (thick black line) of the surrounding drainage basin. Red lines indicate the linear trend calculated from 1979 to 2016. All regression coefficients (β) are statistically significant at $P < 0.001$; **c** comparison between monthly mean regional temperature anomalies, averaged over 13°–17°S − 42.5°–46.5°W, derived from gridded ground-based station data[30] and from CRU TS 4.06 temperature data[72].

water. The study site is located in a protected area under a karst relief characterized by cliffs, areas of exposed epikarst and dolines with steep-sided depressions, which effectively prevents the expansion of farms and agricultural activity in areas surrounding the cave. Therefore, land-use change is unlikely to have played a significant role in affecting the climate surrounding the cave or the geochemical composition of the speleothems. The cave displays a linear plan pattern and a wide-open entrance lying at the bottom of a steep-sided hill. The entrance measures about 50 m in length and nearly 10 m in height,

giving access to the main cave room (Supplementary Fig. S4). The atmospheric conditions at the sampling site are typical of a well-ventilated environment, characterized by large seasonal amplitudes in relative humidity and temperature, reflecting the atmospheric and environmental conditions outside the cave (Supplementary Fig. S5). Variations in Onça Cave relative humidity reflect the relative humidity and evaporative potential from the exterior environment, as documented by the comparison between in-situ measurements inside the cave and data from a nearby local meteorological station

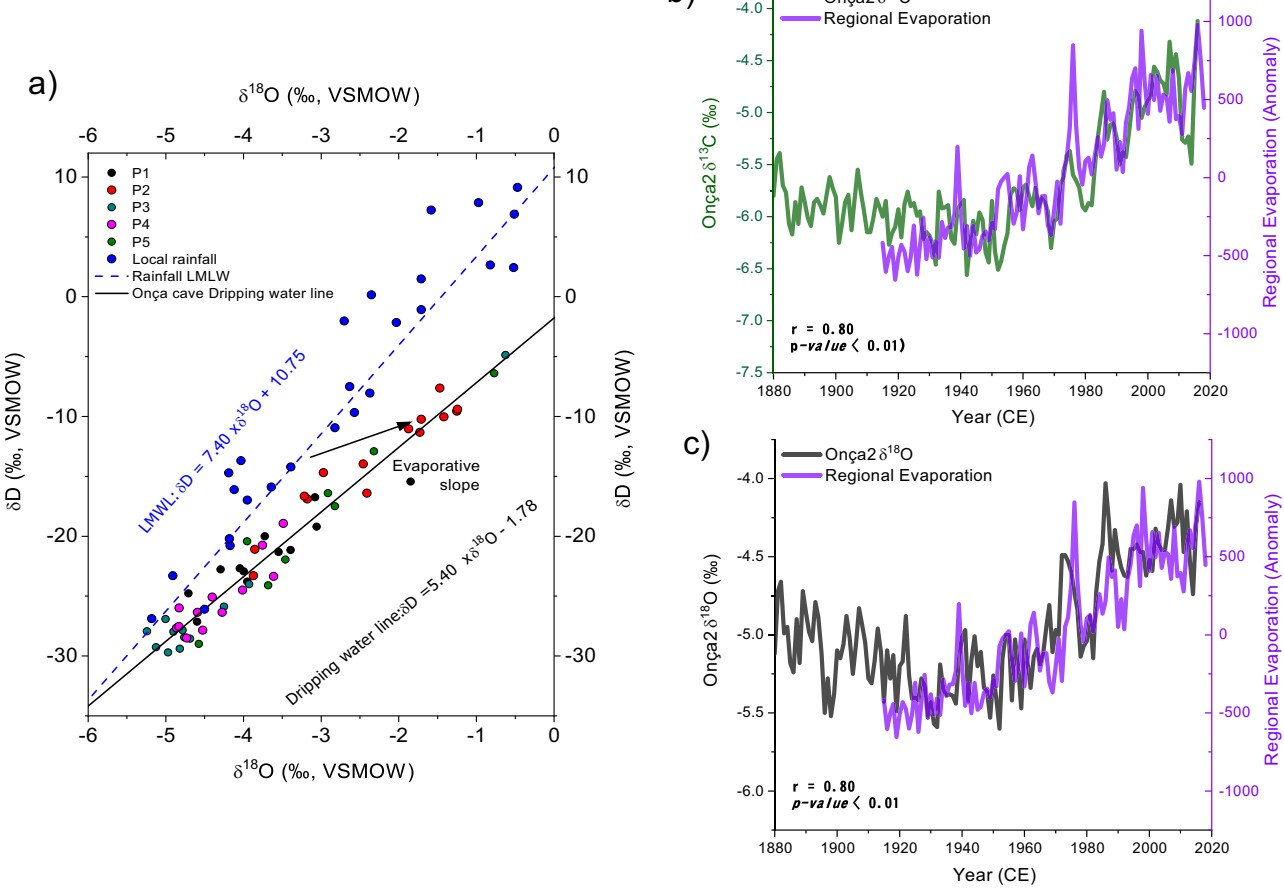

**Fig. 2 | Cave atmosphere evaporative effects on stable isotopes. a** Evaporative effect on isotopes from cave drip water from Onça Cave: Comparison between δ¹⁸O and δD of monthly local rainfall (Januária INMET Station, this study) and cave drip water from five monitoring sites (P1 to P5) (this study). The dashed blue line represents the local meteoric water line (LMWL). The black line represents the drip line from well-ventilated Onça Cave. **b, c** Comparison between δ¹⁸O and δ¹³C from the Onça2 speleothem and regional evaporation (1914–2016) obtained from monthly mean data, averaged over the domain 13°–17° S and 43°–45° W (Supplementary Table S1).

(Supplementary Fig. S5). In the main cave room, the relative humidity varies from ~50 to ~100%, while the cave temperature oscillates between 17 and 25 °C, with low and high values during winter and summer, respectively (Supplementary Fig. S5).

The record from Onça Cave is composed of two speleothems, named Onça2 and Onça4 (Supplementary Fig. S6 and Supplementary Table S2), providing an annually resolved chronology covering the last 718 years spanning from 1298 to 2016 CE. The chronology of Onça2 was constructed using a layer counting method, validated against 19 U-Th dates ranging from 2015 ± 3 to 1760 ± 4 years CE ("Methods" and Fig. S7). Conversely, the chronology of Onça4 was constructed based on U-Th ages, integrating a set of 19 ages ranging from 1852 ± 2 to 1298 ± 3 years CE ("Methods" and Supplementary Fig. S7). To reconstruct long-term changes in the regional hydroclimate and changes in the evaporative demand we combine trace elements ratios (Mg/Ca, Sr/Ca, and Ba/Ca) with oxygen and carbon isotopes. The role played by the evaporative demand in determining variations of the oxygen isotopes was assessed by comparing the δ¹⁸O of speleothems dripping from five different points with the isotope composition of the local rainfall. The comparisons allow us to detect nonequilibrium isotope fractionation related to evaporative effects in the cave environment (see "Methods" and Fig. 2). In addition, we performed an experiment using artificial substrates to analyze the seasonal geochemical effects of temperature and relative moisture on speleothem calcite from Onça cave. Five dripping sites (P1–P5) were monitored and sampled every month from July 2018 to October 2019. The monitored parameters

include temperature, relative humidity, and calcite deposition rate (see "Methods" for details) (Supplementary Fig. S8). In addition, we tested the relationship between the geochemical proxies with instrumental time series from local meteorological stations[33–35] by estimating a scaling factor using a least-squares regression between the geochemical proxies and the instrumental records that permits us to assess the relative contribution of the related environmental forcing to the proxy signal by testing the linear similarity among the time series (see "Methods").

## The effect of PET on the local hydrologic balance

The significant negative trend seen in the local streamflow since the 1980s (Fig. 1b) points to a prolonged hydrologic drought prevailing over central–eastern Brazil. In 2010 the local streamflow dropped to its lowest level since the start of the monitoring around 1940. As shown in Fig. 1, the good correspondence between the negative trend observed in the streamflow of local rivers and the regional precipitation indicates that rainfall plays an important role in contributing to the hydrologic deficit. The persistently low precipitation since the 1990s, has contributed to increased hydrologic stress in recent years. However, precipitation is not the only determining factor for hydrologic drought. Indeed, the negative trend recorded in the z-score values of the streamflow (slope: −0.054 ± 0.007) is comparatively higher than the one observed in the precipitation time series (slope: −0.039 ± 0.013). Conversely, the calculated P-PET index shares nearly the same trend as the streamflow (−0.052 ± 0.011) (Fig. 1b). The

significance of the regression coefficients calculated for each variable are confirmed by the F-test, presenting P values < 0.001. From 1979 to 2016, streamflow of local rivers decreased by 20% per decade in terms of m³/s. During the same time period local precipitation decreased only 7% per decade (70 mm.decade⁻¹) while the P-PET presents a decreasing trend of 18% (125 mm.decade⁻¹), indicating that evapotranspiration changes represent an important component of the current drought. These comparisons suggest that the negative trend observed in the river discharge starting in the 1980s is partially a result of the increased evapotranspiration demand with increasing local temperatures. A recent detection and attribution study from Chagas[5] shows that the reduced streamflow over central–eastern South America is aligned with decreasing P-PET. Furthermore, a strong trend towards aridification over central–eastern South America after 2010 is also consistent with model projections that point to a severe decrease in soil moisture, even under the moderate RCP 4.5 emission scenario, as a consequence of increased evapotranspiration[3,5,36]. Yet, we cannot completely rule out a contribution from land-use changes to the rapid decrease observed in regional streamflow. As pointed out by Chagas[5], the accelerated reduction of streamflow over central–eastern Brazil from the year 2000 onward points to intensive agriculture as a relevant driver of the water deficit.

## Geochemical proxies

The long-term trends in $\delta^{18}O$ and $\delta^{13}C$ recorded in the Onça Cave speleothems bear a close resemblance to the observed increase in aridity documented in the instrumental record (Fig. 2), suggesting increasing temperature and local evaporative conditions as important drivers for the decadal variability recorded in the speleothem geochemical proxies from Onça Cave (Fig. 2c). For oxygen isotopes, temperature and relative humidity represent competing forces that drive values in opposite directions[23,37–39]. In remote and isolated cave chambers, the constant temperature and high and stable relative humidity prevent substantial isotopic changes, allowing the speleothems to provide reliable records of rainfall isotope variability[38,39]. However, an increase in temperature during calcite deposition leads to a reduction of the $\delta^{18}O$ values of the forming calcite due to a reduction of the calcite-water isotope fractionation factor ($\alpha_{calcite-water}$). Empirical measurements point to a rate of approximately −0.2‰/°C[40–42]. However, our $\delta^{18}O$ record presents a trend in the opposite direction, with a ~0.7‰ increase, despite a warming of about 2° since 1970. Conversely, an increase in evaporative conditions also increases the $\delta^{18}O$ values of the calcite formed in speleothems. Thus, the trend seen in the Onça2 $\delta^{18}O$ values after the 1970s is primarily a result of kinetic fractionation related to the highly evaporative conditions and not a direct consequence of temperature variations. This assessment is supported by the isotope monitoring of Onça Cave dripping water, which clearly shows that the $\delta^{18}O$ and $\delta D$ values of the drip water plot on an evaporation line (Fig. 2). As can be observed in Fig. 2a, the $\delta^{18}O_{water}$ and $\delta D_{water}$ values of the dripping solution from Onça Cave define a meteoric water line with a slope of about 5.4, which is significantly less than the meteoric water line of the local rainfall (slope = 7.4), suggesting that evaporation is affecting the isotopologues of the cave dripping water. Under such conditions, the speleothems potentially reproduce the relative variations of the evaporative demand, recording remarkably high positive excursions during periods characterized by a water deficit. The dripping rate also plays an important role in modulating the isotope effects produced by changes in the cave atmosphere environment. For instance, the evaporative effect is strongly controlled by the dripping rate, since the time of re-equilibration between oxygen isotopes and cave atmosphere controls the extent of the isotope fractionation effect[23,43,44]. Thus, as the dripping rate decreases, the longer exchange time between dripping solution and cave atmosphere enhances the evaporative effect on $\delta^{18}O$ and $\delta^{13}C$ values of the speleothem[23].

Finally, the increase in the $\delta^{18}O$ values of the Onça Cave speleothem could also result from a decrease in the local rainfall amount[45–47]. However, the positive trend observed in the $\delta^{18}O$ from Onça2 stalagmite starting around 1970 (Fig. 2) is too high to be explained by rainfall oxygen isotope variability alone. Indeed, estimates of changes in rainfall oxygen isotope values based on monitoring studies[47] document a positive $\delta^{18}O$ trend of 0.007 ± 0.003 ‰.yr⁻¹. Meanwhile, the Onça2 stalagmite shows a trend that is twice as large (0.015 ± 0.002 ‰.yr⁻¹) (Supplementary Fig. S9).

In general, the trace elements also share a common trend with the isotope record, characterized by an increase in the elemental ratio values since the 1970s (Fig. 3). These trends are opposite to the P-PET (Fig. 1a) and in phase with evaporation trends (Fig. 2b), which suggests a close link between decreased water recharge and processes like the PCP controlling the geochemical composition of the Onça Cave speleothems. How the cave atmospheric conditions determine the evaporative forcing and thus drive proxy variability can also be observed in the results from the calcite deposition experiment carried out at Onça Cave (Supplementary Fig. S8). In general, Mg/Ca ratios in farmed calcite vary out of phase with cave relative humidity, i.e., high values occur during periods of reduced relative humidity in the cave (Supplementary Fig. S8). These results confirm the PCP as the driver of trace element variability.

The combined effect of reduced rainfall and exacerbated evaporation lead to a decrease in the meteoric water recharge that may favor the PCP in response to decreasing water levels along the fractures and conduits of the cave vadose zone, which is reflected as higher Mg/Ca in the speleothems[29,48–50]. The role of the pluvial recharge in driving trace element variability in the Onça speleothems becomes apparent when comparing the detrended time series of Mg/Ca from Onça2 with local precipitation (Supplementary Fig. S10). The two variables are negatively correlated with an antiphased decadal variability, which reinforces the concept of hydrologic recharge leading to PCP as the main driving force of trace element variability (Supplementary Fig. S10). Nevertheless, a further increase in the trace element values can be attributed to the increase in the potential evapotranspiration that significantly reduced the meteoric water recharge in the last 40–50 years (Fig. 3). As we demonstrate in Fig. 3, variations in the trace elements from the Onça stalagmite are consistent with changes in evaporative demand, which ultimately, are driven by rising temperature (Fig. 1). To assess the correspondence between the observed trend in isotope and trace element ratios with the observed contemporaneous change in temperature, precipitation, and the evaporative demand, in Fig. 3 we regressed the z-scored time series of $\delta^{18}O$, $\delta^{13}C$, and trace element ratios against z-scored time series of these suspected climatological drivers using an ordinary least square regression. The significance of the regression coefficients calculated for each variable are confirmed by the F-test, presenting P values < 0.001. For this test we relied on the same change point detection algorithm by Killick[31] to first determine the onset of the common trend (Supplementary Fig. S2), which occurs around 1942 (see "Methods"). We then compared this trend with the long-term temperature record from CRU TS 4.06 (Fig. 1c). To extend the PET to 1942, we used the Thornthwaite (1948) equation but applied a monthly coefficient correction factor that allows for an approximation to the PET based on the Penman–Monteith equation[51].

The scaling factor calculated using ordinary least square regression highlights that the $\delta^{18}O$, $\delta^{13}C$, and trace element ratios share similar trends with the evaporative demand, (P-PET or evaporation), at multidecadal timescales, which ultimately is temperature-dependent. Based on the comparison of the linear trend, we can state that roughly 70% of the increase in these geochemical ratios can be attributed to the increase in the evaporative demand alone. This analysis is based on z-score data and the scaling factors thus represent units of standard deviation. On the other hand, temperature may drive significant

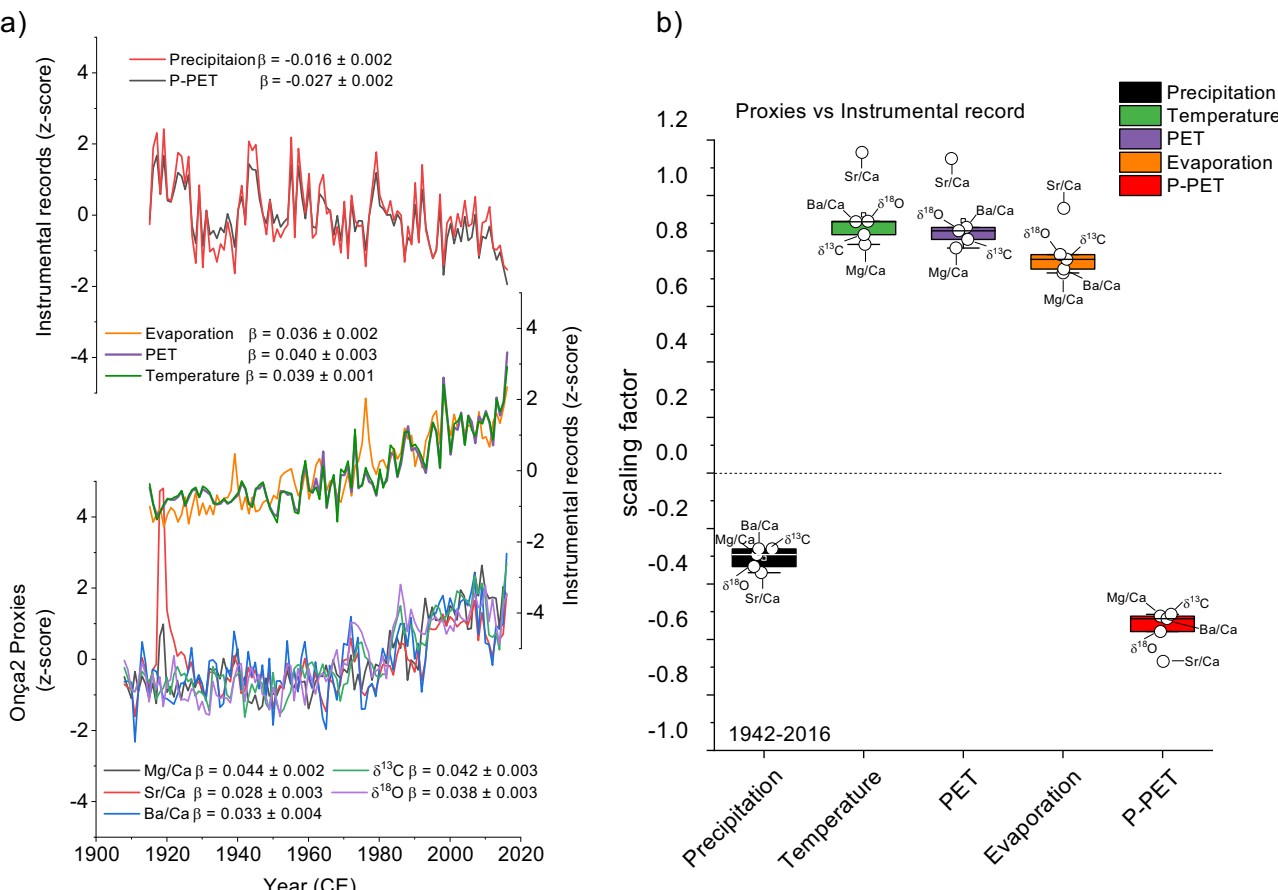

**Fig. 3 | Comparison between geochemical proxies and environmental drivers derived from local meteorological stations. a** comparison between z-scored (between 1915 and 2016) data from geochemical proxies (Mg/Ca, Sr/Ca, Ba/Ca, δ¹⁸O, and δ¹³C) with instrumental records of regional temperature, precipitation, potential evaporation, potential evapotranspiration (PET) and P-PET index. **b** Whisker box plots of regression coefficients between geochemical proxies and main environmental drivers between 1942 and 2014 using a smoothed 5-year running mean: precipitation (black), temperature (green), potential evapotranspiration (purple), potential evaporation (orange) and precipitation potential

evapotranspiration (red). The slopes of the proxy time series trends after 1942 (determined using a regime-shift test) are represented by regression coefficients (β) and are statistically significant at $P < 0.001$. Temperature data are derived from (CRU TS 4.06 at 13°–17°S; 42.5°–46.5°W), precipitation and evaporative potential are derived from INMET and ANA meteorological stations shown in Fig. 1. Potential evapotranspiration was calculated using the Thornthwaite (1948) equation from temperature data and adjusted to Penman–Monteith (FAO-56) using monthly coefficient corrections according to Aschonitis[51].

variations in Mg/Ca in speleothems through changes in the partitioning of Mg uptake into calcite, leading to high Mg/Ca values in warm climates[52–54]. However, variations in Mg/Ca in the dripping solution, mainly associated with PCP and incongruent dissolution of dolomite, override the effects of temperature by more than an order of magnitude[54].

## Climate implications

Simulations from the Coupled Model Intercomparison Project Phase 6 (CMIP6)[55] show that global temperature trends since the mid-1970s cannot be reproduced in climate models without considering anthropogenic forcing. Similarly, the timing of the observed increase in aridity recorded in the speleothem from central–eastern Brazil, starting in the 1970s, suggests an anthropogenic contribution (Fig. 4). To investigate the main drivers of changes in the regional hydrologic balance and to assess the anthropogenic contribution, we perform a Detection and Attribution (D&A) analysis[33–35]. The method is based on the estimation of scaling factors (regression coefficients) obtained from the regression between the geochemical proxies Mg/Ca and δ¹⁸O values of the speleothem and PET and P-PET simulated by an ensemble of CMIP6 models from the Detection and Attribution Model Intercomparison Project (DAMIP) (see "Methods"). The test was performed using data from natural and anthropogenically forced historical

simulations (1850–2014 CE) covering a region of 4° × 4° (13°–17° S; 42.5°–46.5° W) centered over the cave site; the same region used when calculating the hydrologic balance using instrumental records. For this purpose, we selected three different experiments: (i) a historical experiment (GHG + NAT) that includes natural forcing represented by time-evolving externally imposed volcanic aerosol and solar irradiance variability, as well as anthropogenic greenhouse gas and aerosol forcings; (ii) a natural historical experiment (NAT)–this simulation is forced only with solar irradiance and stratospheric aerosol forcing from volcanic eruptions, and (iii) a historical anthropogenic forcing experiment (GHG)–forced with the well-mixed greenhouse-gas changes only[34]. Our analysis is based on a multi-model ensemble, composed of 8 different CMIP6 earth system models (Supplementary Table S3). The times series were smoothed using a 5-year running mean in order to focus on the multiannual variability.

The relative contribution of the individual forcing components to the increased aridity observed in the Mg/Ca and δ¹⁸O speleothem records can be assessed in the whisker plots and in the time series for PET and P-PET calculated based on temperature and precipitation data extracted from the ensemble of the different model experiments (Fig. 4). The scaling factors, represented by the beta coefficient, calculated based on the z-score data indicate the extent to which the trends in Mg/Ca and δ¹⁸O values are explained by the individual

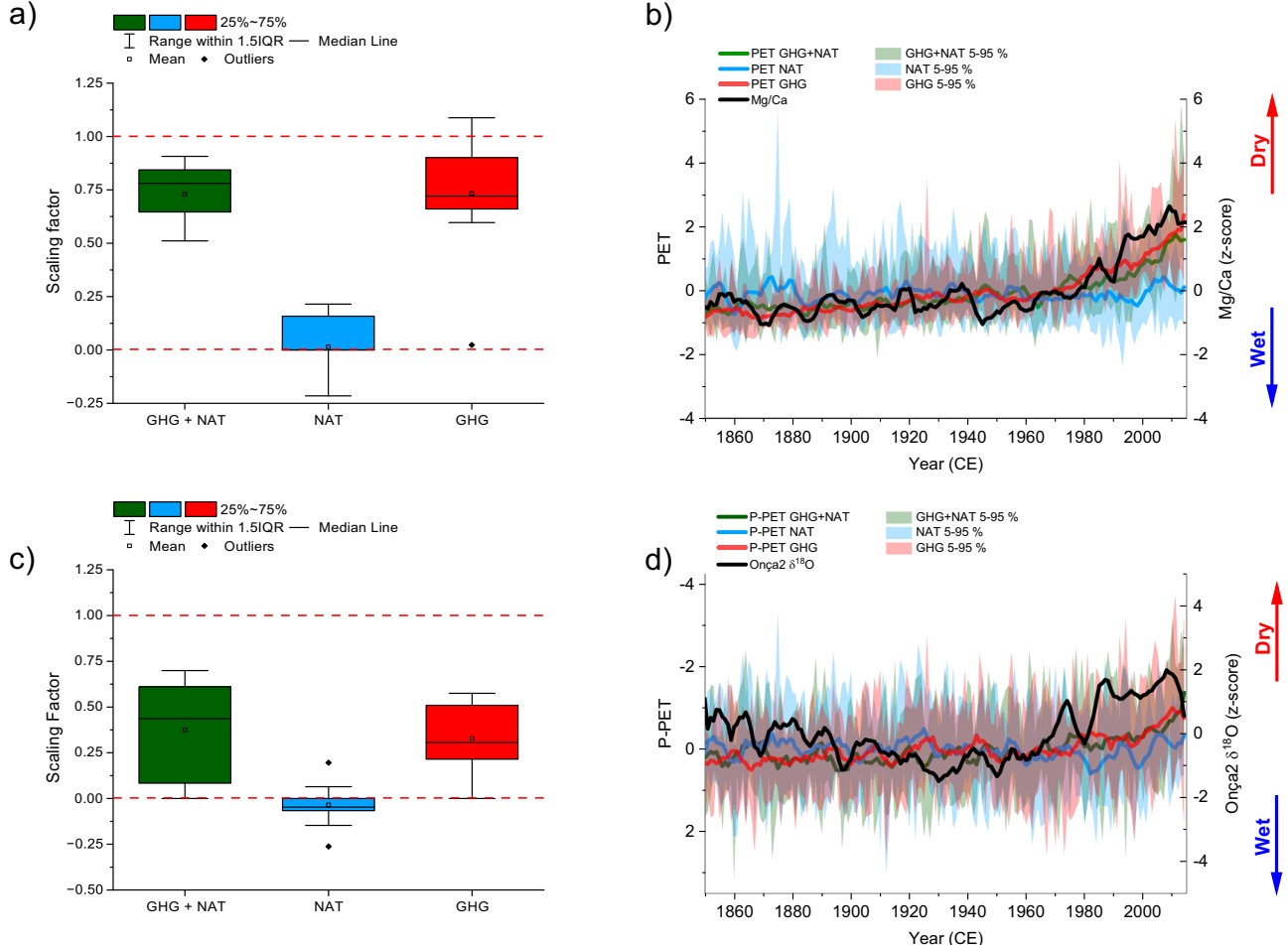

**Fig. 4 | Detection and attribution analysis. a** Scaling factors for simulated ensemble (PET) versus speleothem Mg/Ca in natural and anthropogenic greenhouse gas (GHG + NAT), natural forcings only (NAT) and anthropogenic greenhouse gas only (GHG), experiments from DAMIP/CMIP6 earth system models. Lower and upper box boundaries determine the 25th and 75th percentiles, respectively, while the lower and upper whisker boundaries determine the 5th and 95th percentiles. **b** Five-year running mean simulated PET time series from DAMIP/ CMIP6 ensemble mean (solid lines) for natural and anthropogenic greenhouse gas (GHG + NAT), natural forcings only (NAT) and anthropogenic greenhouse-gas only (GHG) experiments. The 5–95% range of the model spread is presented as shaded areas; **c, d** present similar tests as shown in (**a, b**), respectively, but comparing speleothem $\delta^{18}O$ record with simulated hydrologic balance P-PET calculated from DAMIP/CMIP6 experiments. Source data are provided as a Source Data file.

forcing. They show the magnitude by which the dependent variable (Mg/Ca and $\delta^{18}O$) changes in response to a one-unit increase in the predictor variable (forcing). The distribution of the beta coefficients across the eight models for both the GHG + NAT and GHG simulations are significantly different from zero, indicating a detectable influence of anthropogenic and combined natural-anthropogenic forcing on Mg/Ca and $\delta^{18}O$ in the simulations. On the other hand, the scaling factors calculated using natural forcing simulations only (NAT) are not significantly different from zero, regardless of the variable considered. These results show that Mg/Ca and $\delta^{18}O$ at our cave site are responding to anthropogenic climate change, and that natural forcing alone cannot explain the observed trends in the speleothem records. The spread of the scaling factors obtained in the comparison between $\delta^{18}O$ and P-PET derives from the differences in the simulated precipitation across the models. In general, simulating precipitation at such a regional scale is much more challenging for climate models than simulating temperature. Problems related to boundary conditions and misrepresentation of physical processes associated with precipitation-forming mechanisms result in significant inconsistencies between simulated and real precipitation, particularly in convective environments and on local scales[56,57]. However, the trend expressed by Mg/Ca and $\delta^{18}O$ and the increased aridity starting around the 1970s is

consistent only with the forcing in the anthropogenically forced experiments (GHG + NAT and GHG), which confirms the notion of a major influence of the anthropogenically induced temperature increase on the potential evapotranspiration as recorded in our speleothems (Fig. 4).

The role of temperature in driving hydroclimate variability in this region is evident in the 20th century, but the lack of longer and precisely dated and high-resolution temperature records covering eastern South America complicates a comparison prior to the 20th century. To circumvent this issue, we compared the regional hydroclimate variability recorded in the speleothems from central–eastern South America with the gridded surface temperature reconstruction from the PHYDA product[32].

The most prominent feature recorded in the Onça2 stalagmite is the anomalous increase in the Mg/Ca values pointing to a prominent drying trend after 1970, which is consistent with the other geochemical proxies such as the $\delta^{18}O$ and $\delta^{13}C$ records (Fig. 5). Considering just the Onça2 stalagmite, it is readily apparent that the Mg/Ca, $\delta^{18}O$, and $\delta^{13}C$ values over the last decades have no counterpart since 1729. Moreover, the combined $\delta^{18}O$ and $\delta^{13}C$ records from the Onça2 and Onça4 stalagmites show that the drying trend observed from the 1970´s onward is unprecedented in the past 720 years (Fig. 5). Notably,

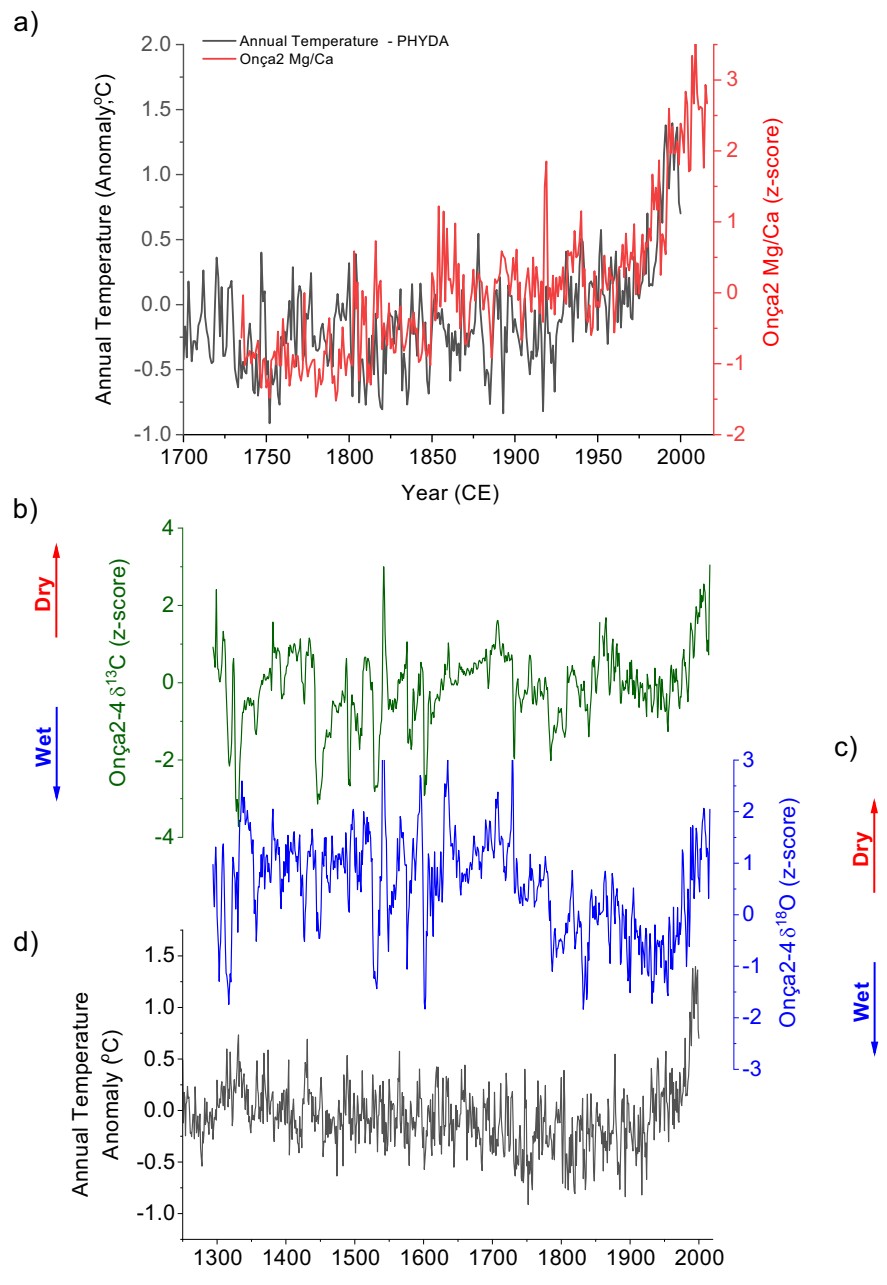

**Fig. 5 | Onça speleothem record vs. reconstructed annual mean temperature.**
**a** comparison between the ensemble mean annual surface temperature anomaly for 12°–16° S to 41°–47° W from PHYDA with Mg/Ca from Onça2 speleothem; **b**–**d** same as in the (**a**) but using δ¹³C and δ¹⁸O respectively, from the composite from Onça2 and Onça4 speleothems. High $\delta^{13}C$, $\delta^{18}O$, and Mg/Ca values indicate highly evaporative conditions, driven by reduced recharge, lower cave relative humidity and warmer temperatures.

the observed drying trend is clearly detached from the decadal variability prior to the 20th century. From 1300 to 1850 CE the isotopic excursions are characterized by abrupt short-lived departures from the mean state, lasting ~10 years. Those events have no counterpart with the climate variability of last two-centuries and may be associated with abrupt climate changes related to variations in external forcing, such as large volcanic eruptions or solar irradiance changes. Conversely, in the 20th century, the isotopic variability is superimposed on a long-term isotopic enrichment. This notion is supported by results from a wavelet analysis, which indicates that multidecadal variability with a periodicity of ~60 years dominated the record from 1300 to 1800 CE (Supplementary Fig. S11). When combined, the two speleothem records from Onça Cave reinforce the idea that temperature

is playing a major role in the hydrological balance of central–eastern Brazil during the last decades by modulating the potential evapotranspiration, thereby further exacerbating the drought conditions in the region.

In conclusion, the speleothem record from central–eastern Brazil points to an increased risk of future drought in the savannah biomes, primarily forced by the evaporation trend driven by anthropogenic global warming. The unprecedented upward trend seen in the geochemical proxies results from a combination of factors, including surface warming, increasing evaporative potential, and reduced precipitation, reflecting environmental conditions not seen in at least the past 720 years. The risk of an exacerbated future hydrologic drought is likely, given the role played by GHG forcing as shown in our study. This

risk may be further exacerbated by the projected enhanced rainfall seasonality (e.g., Liang[58]), which will enhance the water deficit during the long dry season. Biomes such as the tropical savannah, where the dry season usually lasts five months or more, and which are home to vast areas of crops, native forests and cities with large populations, will therefore be at severe risk of long-term future drought. Our results also point toward temperature as an important factor to consider in the hydrological balance over the tropics, due to its ability to exacerbate droughts[13,59,60].

## Methods

### Speleothem chronology
The Onça2 stalagmite has a robust chronologic control, based on 19 U-Th ages, covering the time period from 1760 ± 4 to 2015 ± 3 years CE, with an age uncertainty varying between 1 and 5 years (Supplementary Figs. S6 and S7). The Onça2 stalagmite also shows an annual layer structure that consists of sub-annual pairs of light and dark laminae, that enabled us to apply layer counting as an additional chronologic control. The accuracy of the chronology based on layer counting was assessed in comparison with the U-Th ages by applying the StalAge algorithm, which uses a Monte-Carlo simulation to construct an age model based on the best adjustment of linear trends through sub-sets of three adjacent data points[61]. In general, the chronology based on layer counting is in good agreement with the U-Th dates along the speleothem growth axis. Overall, the speleothem includes 255 layers, consistent with the time interval obtained from the U-Th ages (U-Th sample On2-0.1 = 2015 ± 3 years and On2-17.8 = 1760 ± 4 years CE), defining a temporal range of 256 ± 7 years, Supplementary Fig. S7 and Supplementary Table S2). The Onça4 stalagmite record is composed of 22 U-Th ages, spanning from 1298 ± 2 to 1852 ± 2 years CE (Supplementary Fig. S7). The chronology of Onça4 was constructed using the StalAge algorithm optimal adjustment of linear trends between ages using a Monte-Carlo simulation[61]. The results of the U-Th ages are presented in Supplementary Table S2. The dating work was performed at the Institute of Global Environmental Change, Xi'an Jiaotong University, Xi'an, China.

### Streamflow data
We selected a total of 13 fluviometric stations from ANA (Supplementary Table S1, see also data availability) covering the main tributaries of the São Francisco River in a 4° × 4° area (13°–17° S; 43°–47° W) centered at the cave site. The limited number of stations before 1970 prevents us from assessing the spatial coherence of the overall fluviometric record. Therefore, we limit our analysis to the time period between 1970 and 2018 (Supplementary Table S1). The fluviometric stations were selected according to the following criteria: (i) at least 25 years of data, with less than 5% of missing data; (ii) with data starting before 1990; and (iii) with data ending after 2010. To calculate the mean streamflow, we first standardized each annual time series (z-score) between 1976 and 2016 before creating a regional average based on the median during the hydrologic year (from September to August). Since we are interested in analyzing the effect of natural forcings, i.e., precipitation and evaporation, we removed the record from the São Francisco River, which is strongly affected by urban occupation and water reservoirs.

### Iscam
To construct a single composite for the isotope and trace element records, we applied the iscam algorithm from Fohlmeister[62]. This age-depth modeling software uses a Monte-Carlo approach on absolute age determinations to find the best correlation between different climate proxies from adjacent archives. The iscam method was applied to the composites of $\delta^{18}O$ and trace elements from the Onça2 and Onça4 stalagmites (Supplementary Fig. S12). Age uncertainties at 68%, 95%, and 99% significance levels are obtained from an evaluation of a

set of 2000 first-order autoregressive processes (AR1) for each record, which have the same statistical characteristics as the individual records. This method allows significantly reducing the age uncertainty within the overlapping periods and it can be tested if the signal of interest is indeed similar in all the records[62].

### Carbonate oxygen and carbon isotopes
The oxygen and carbon stable isotope analysis of speleothem calcite was performed at the Stable Isotope Laboratory at the University of São Paulo. Oxygen and carbon isotope ratios are expressed in $\delta$ notation, the per mil deviation from the VPDB standard according to the following equations:

$$\delta^{18}O = [((^{18}O/^{16}O)_{sample}/(^{18}O/^{16}O)_{VPDB}) - 1] \times 1000 \qquad (1)$$

$$\delta^{13}O = [((^{13}C/^{12}C)_{sample}/(^{13}C/^{12}C)_{VPDB}) - 1] \times 1000 \qquad (2)$$

The collected calcite powder was analyzed with an online, automated carbonate preparation system linked to a Finnigan Delta Plus Advantage mass spectrometer. Samples were collected along the growth axis of the speleothem with a sampling resolution varying from 0.125 to 0.4 mm. This sampling approach provided a near-annual resolution.

### Water oxygen and hydrogen isotopes
The analyses of $\delta^{18}O$ and $\delta D$ from cave dripping water and rainwater were performed at the Centro de Pesquisas de Águas Subterrâneas at the University of São Paulo (IGc-USP) using a Laser absorption spectrophotometer of the brand PICARRO L2130i. Data were processed by Laboratory Information Management System (LIMS) for Lasers software[63,64]. Values are reported with an analytical precision of 0.09‰ for $\delta^{18}O$ and 0.9‰ for $\delta D$ relative to Vienna Standard Mean Ocean Water.

The monitoring of $\delta^{18}O$ and $\delta D$ at the Onça Cave dripping sites started in February 2018 and continued until November 2019. A total of five dripping sites for monthly sampling were installed. The water samples were manually collected and stored in 8 ml high-density polypropylene bottles. To avoid isotope equilibration between water and atmosphere the flasks were completely filled and stored in a refrigerator at 5 °C.

A local rainfall isotope monitoring study was performed in the region from November of 2011 to October of 2017. The isotope monitoring station, here referred as "Januária-INMET Station" (15°26'53"S; 44°21'59"W) is located about 40 km south of the cave site, at the INMET meteorological station (OMM: 83386) in Januária City. Local rainfall was collected weekly using a modified version of the rainfall water collector developed by Gröning[65]. The device was improved here by including one more layer of heat dissipation. These samples were collected in 8–30 mL high-density polypropylene bottles. Monthly rainfall $\delta^{18}O$ values for local precipitation (Supplementary Fig. S9) were calculated based on the weighted mean of total precipitation between samplings.

### Detection and attribution analysis
Detection and Attribution (D&A) analysis was performed using z-scored time series according to the following equation:

$$z-score = (x_i - \mu)/\sigma \qquad (3)$$

where $x_i$ is data at the time i, $\mu$ is the mean value, and $\sigma$ is the standard deviation of the time series. For this analysis we used a 5-year running mean time series which mitigates the influence of shorter-term variability and highlights the underlying multidecadal trends. The ensemble of each experiment was calculated using the median of the

eight models. To overcome the problems related to inter-dependency among observations (non-independent and non-identically distributed data) and retain the multidecadal signal and the long-term trend we performed linear regressions using 20 Monte-Carlo simulations based on the minimax algorithm from Xu and Xiaihua[66] (Supplementary Figs. S13 and S14). In this approach, no white noise filtering was applied. For the regressions between speleothem geochemical proxies and CMIP6 simulations we used the period between 1900 and 2015. The resulting regression coefficients (β) provide a scaling factor that is used to compare the trends between time series. A standard t-test was applied to test whether the regression coefficients were significantly different from one another at the 95% level (Supplementary Table S4)[67].

This analysis shows that the scaling factor for the NAT scenario is significantly different from the GHG and the GHG + NAT scenario, and that we cannot reject the hypothesis that the scaling factor for NAT equals zero, while the scaling factors for GHG and GHG + NAT are distinct from zero. The proxies, i.e. δ[18]O or Mg/Ca were used as the response variable (x) and the different forcing simulations (e.g., temperature in the GHG simulation) were used as the explanatory variable (y). To obtain values of PET from CMIP6 runs we used the annual mean temperature output. Absolute values of PET were then calculated using the Thornthwaite (1948) equation and the monthly coefficient from Aschonities[51] that allows an approximation to the PET based on the Penman–Monteith equation.

### Change point detection algorithm
To identify the exact point where the trend along a time series changes, we performed a regime-shift test using a change point detection algorithm[31,68]. The tests was applied to the streamflow and speleothem proxy time series (Supplementary Fig. S2). In the streamflow time series, we assumed two change points, based on a visual inspection, given the two clear peaks around 1980 and 2004 (Supplementary Fig. S2). The algorithm identified two change points around the times 1979 and 2004, separating three periods with different trends; the first trend representing a flat and stable streamflow while the second and the third trend show a consistent decrease toward lower values. After determining the desired number of change points, the algorithm proceeds iteratively by randomly selecting points in the time series. The final estimate is obtained by choosing points where the sum of the squared errors is minimized. In other words, for each desired number of change points (n), the algorithm iteratively defines n + 1 trends in each iteration and compares the sum of squared differences between the signal values and the least-squares fit (error sum of squares). The combination of n points that minimizes the error sum of squares represents the best estimate for change point detection[31,68]. To detect changes in a trend, the F-test is employed on the linear coefficients obtained pairwise along the time series.

### Calcite monitoring
The calcite monitoring was performed using concave glass dishes of 11 cm diameter, also known as watch glass. Each watch glass was placed under the monitored dripping site. The experiment was replaced during every cave trip from July 2018 to October 2019, resulting in a nearly monthly sampling resolution. To monitor the calcite deposition, the watch glasses were weighed prior to deployment and after collection, yielding a mass of calcite of about 200 mg month$^{-1}$. A common sampling surface with a radius equal to 5 cm around the center of deposition was defined between the glasses. The calcite was collected through manual scraping using a scalpel.

The calcite sample preparation was performed by weighing 10 mg of the homogeneous sample and adding 5 mL of 2%(v/v) HNO$_3$ solution, which was stirred and filtered with 0.22-µm PVDF filter. Each sample was diluted 100 times for Ca quantification, so 100 µL of each solution was taken for a total volume of 10 mL (in HNO$_3$ 2% v/v/ solution). Quench water samples were collected at the sampling site,

filtered, and acidified to 2% v/v HNO$_3$. The elemental quantification of Ca (315.8 nm) and Mg (279.5 and 280.2 nm) were analyzed via ICP-OES (iCAP 6000 Series, Thermo Scientific, Bremen, Germany), in radial view, in partnership with the Spectrometry, Sample Preparation and Mechanization Group (GEPAM) at the Institute of Chemistry (University of Campinas, Brazil, SP) coordinated by Prof. Dr. Marco A. Z. Arruda.

### Trace elements
Mg/Ca, Ba/Ca, and Sr/Ca ratios in the Onça stalagmite were obtained using Cavity Ring-Down Spectroscopy (CRDS) at the Laboratorio de Estudios Isotópicos at the Centro de Geociencias of the Universidad Nacional Autónoma de México (UNAM) following the methodology[29] using a Resonetics M-50 laser ablation workstation with a large-format two-volume ablation chamber previously described in ref. [69] and ref. [70]. We projected a 40 µm ArF excimer laser (193 nm, 20 ns FWHM) onto the sample surface with a fluence of 3 J/cm$^2$ and a 5 Hz frequency. The laser was rastered along the growth axis at a speed of 0.5 mm/min within a He atmosphere, allowing to obtain a nearly continuous record of elemental variability in the stalagmite with sub-annual resolution. The resulting aerosol was admixed with Ar and transported to an iCap-Q ICP-MS where signal intensity ratios were measured. The analysis of the samples was bracketed by the analysis of SRM NIST 612, which allowed us to calculate elemental ratios (mmol/mol) using the reported composition by Jochum et al.[71] or the NIST glasses.

Analyses of Mg/Ca ratios from calcite deposited on glass substrates were performed in the Department of Chemistry of the Universidade de Campinas using ICP-OES. Near 10 mg of calcite powder was sampled from glass substrates and diluted in 15 ml of HNO$_3$ 2%. The relative standard deviation between the measurement levels yields an analytical precision of ~1.1% for Ca and 0.74% for Mg.

## Data availability
The speleothem U-Th ages generated in this study are provided in the Supplementary information. The geochemical records, including δ[18]O, δ[13]C and trace elements records from Onça2 and Onça4 stalagmites and composite from Onça2 and Onça4 stalagmites are provided in Supplementary Data 1.xls. The instrumental records for regional precipitation, evaporation, potential evapotranspiration and streamflow are presented in the Supplementary Data 2.xls file. The data of potential evaporation calculated to 13–17°S; 42.5–46.5°W were obtained from Brazilian Daily Weather Gridded from Xavier[30] and are available at https://sites.google.com/site/alexandrecandidoxavierufes/brazilian-daily-weather-gridded-data? authuser=0. The streamflow data are also available at http://www.snirh.gov.br/hidroweb/. The cave monitoring data generated in this study, including isotope data from cave drippings, modern calcite deposition rate and Mg/Ca ratio from farmed calcite and cave temperature and relative humidity are presented in the Supplementary Data 3 (tabs: Cave dripping isotope data; Monitoring Mg_Ca and dep rate; Cave temp. and RH). Local rainfall isotope data from Januária Station generated in this study, including the records from GNIP Station are presented in Supplementary Data 4. The rainfall isotope data from GNIP station: Belo Horizonte CDTN and Brasília Airport are also available at https://www.iaea.org/services/networks/gnip. The surface temperature reconstruction from PHYDA is presented in the in the Supplementary Data 5. The Potential evapotranspiration (PET) and hydrologic balance (P-PET) calculated from CMIP6 experiments GHG + NAT, GHG and NAT are presented in the Supplementary Data 6 (tabs: "CMIP6_GHG + NAT", CMIP6_GHG and "CMIP6_NAT", respectively). Source data are provided with this paper.

## Code availability
The "minmax algorithm" was performed using MATLAB (R2021a). The MATLAB function used in the minmax algorithm is provided in the Source Data file.

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

## Acknowledgements

We are grateful to the *Centro Nacional de Pesquisa e Conservação de Cavernas* (CECAV/ICMBio) for providing permission to collect stalagmite samples. This study was financed in part by the São Paulo Research Foundation - FAPESP (grant numbers: PIRE-CREATE project 2017/50085-3) to F.W.C. and NMS Rio de Janeiro Research Foundation - FAPERJ (grants E-26-201.421-2021 and E-26211.352-2021) to N.M.S. Conselho Nacional de Desenvolvimento Científico e Tecnológico - CNPq (grant 312343/2022-1) to N.M.S; (grant 309755/2022-0) to F.W.C.. National Science Foundation -  NSF (PIRE-CREATE project OISE-1743738) to M.V.; (grant AGS-2202772) to R.R.F. Fundação de Amparo à Pesquisa do Estado de Goiás (FAPEG) (grant number 202110267000878) to P.F.M.B. Marie Skłodowska-Curie Action (ITHACA-101024389) to E.T.. PAPIIT-UNAM IG100722 to J.P.B. VFN has been funded by the Deutsche Forschungsgemeinschaft (DFG, German Research Foundation)—project no. 395588486. National Natural Science Foundation of China - NSFC (grant NSFC 42150710534) to H.C. FONDECYT, PERU (grant PROCIENCIA 124-2020) and Young team associated - French National Research Institute for Sustainable Development - JEAI-IRD (grant CHARISMA Project - JEOECCHARI) to J.A.

## Author contributions

N.M.S., P.F.S.M.B., F.W.C., H.R.S., and M.S.S. conducted the fieldwork. N.M.S., F.W.C., and M.V. conceptualized this study. N.M.S., P.F.S.M.B., E.T., J.P.B., M.T.K., M.V.C., A.A., M.H.S., W.D., D.S.F., M.A.Z.A., R.R.F., and M.S.S. carried out the experiments and data analyses. D.S.F., M.A.Z.A., and J.P.B. contributed to trace element analysis. H.C. and R.L.E. contributed to the U-Th dating work. P.F.S.M.B., H.R.S., and M.S.S. helped organize fieldwork and sampling. G.S., A.A., M.H.S., M.S.S., and M.T.K. contributed to climate analysis. M.T.K., F.W.C., J.A., and V.F.N. interpreted results. N.M.S., M.V., and F.W.C. accomplished the writing with the help of all co-authors. J.L.C. and E.T. conducted statistical analyses.

## Competing interests

## Additional information

[1]Instituto de Geociências, Universidade de São Paulo (USP), São Paulo, São Paulo-SP, Brazil. [2]Departamento de Geoquímica, Universidade Federal Fluminense (UFF), Niterói, Rio de Janeiro-RJ, Brazil. [3]Universidade Estadual de Goias (UEG), Iporá, Goiás-GO, Brazil. [4]Instituto Federal Goiano, Ceres, Goiás-GO, Brazil. [5]Centro de Geociencias, Universidad Nacional Autónoma de México, Querétaro, México. [6]Department of Atmospheric and Environmental Sciences, University at Albany, State University of New York, Albany, NY, USA. [7]Department of Geology, National Museum of Natural Sciences-Spanish National Research Council (MNCN-CSIC), Madrid, Spain. [8]General Coordination of Earth Sciences, National Institute for Space Research (INPE), São José dos Campos, São Paulo, Brazil. [9]Institute of Global Environmental Change, Xi'an Jiaotong University, Xi'an, China. [10]Instituto Federal de Educação, Ciência e Tecnologia do Norte de Minas Gerais, Januária, Brazil. [11]Instituto Geofísico del Perú, Lima, Peru. [12]Universidad Nacional Agraria La Molina, Programa de Maestria en Recursos Hídricos, Lima, Peru. [13]Department of Earth and Planetary Sciences, Harvard University, Cambridge, MA, USA. [14]Department of Earth Sciences, University of Minnesota, Minneapolis, MN 55455, USA. [15]Department of Climate and Space Science and Engineering, University of Michigan, Ann Arbor, MI, USA. [16]Institute of Chemistry, University of Campinas, Campinas, SP, Brazil. [17]Department of Geosciences, University of Tübingen, Tübingen, Germany. ✉e-mail: strikis@usp.br

