## [Peer Review File · Nature Communications]

Modern anthropogenic drought in Central Brazil
unprecedented during last 700 yearsREVIEWER COMMENTS

Reviewer #1 (Remarks to the Author):

Strikis et al. present a new multiproxy record of rainfall and evaporation from stable isotope ($d18O$, $d13C$) and trace element (Mg/Ca, Sr/Ca, Ba/Ca) measurements from two speleothems located in central-eastern Brazil. Overall, the paper is quite well-written and structured, and the conclusions are well thought out. The record itself is also of high quality. The problem, however, is that many assumptions are made re: the controls on the speleothem proxies. Specifically, attempts by the authors to quantify P-PET using speleothem proxies that may have different temporal resolutions to observations and model simulations. In addition, efforts to calibrate (or assess 'linear congruence') the speleothem proxies using observations (via linear regression) can pose problems when factoring in the inherent seasonal biases associated with each proxy. For example, the $d18O$ may be more sensitive to summer monsoon rainfall while the interannual trace element variability may be explained by dry season hydroclimate. These issues should be addressed in the manuscript.

Comments:

(1) The calibration of the geochemical proxies (which have not been shown to be recording annual/seasonal variability) against climate model output using linear regression. Aside from the fact that the different geochemical proxies have their own seasonal biases, the authors make little mention of the fact that the speleothem proxies may be (likely) recording multi-year variability given karst mixing. Hence, conducting a linear regression of this type from data of (likely) varying resolution is problematic and warrants further discussion in the manuscript.

(2) The outdated equation used to calculate PET (using both observations and model output) without consideration of its uncertainties given that it relies heavily on temperature alone. It is well known that evaporative flux is also influenced by factors other than temperature, such as wind speed. Why did the authors only use this rather outdated equation to calculate PET and not test other more updated equations? By doing so may have an influence on their findings. There is a rich body of work from the hydrology community looking at global and regional changes in PET from, for example, satellite data and using more updated equations. A deeper dive into this literature /research may benefit the study.

For a clarification of these points and others, please refer to the annotated pdf-version of the manuscript.

Reviewer #2 (Remarks to the Author):

This paper presents an analysis of speleothem records in central Brazil that the authors argue relate to

changes in the watercycle in the region. The authors conduct a careful analysis of the role of precipitation and evaporation in the data and reconstruct a changing ability to meet evaporative demand from the data. A detection and attribution analysis then concludes that the driver of this change is anthropogenic. Overall it looks like there are some interesting findings here, but the approaches could be explained more clearly and i have some concerns about what exactly was done (which isnt everywhere clear either).

thus i have the following comments

1) i like the careful analysis of the role of the various contributors to hydrological changes and the use of multiple proxies. i dont quite understand why the individual proxies (13O and 18O) have such strong variability while the combined anomaly which is interpreted as temperature anomaly (figure 5) has very robust and consistent variability. can this be more clearly explained? and is the detection attribution analysis conducted on the components - it seems like this from figure 4 - which is interpreted as P-PET. what is the uncertainty in the calibration of one to the other - does this need to be considered? I doubt that the variability in the individual components is consistent with climate model variability.

2) the authors find a change in hydrological variables that is only reproduced if all forcing simulations are used to compare to the data. This is quite convincing as done, but the details are slim in the ms on the attribution approach - has a residual consistency test been conducted? (figure 5 suggests very strange variability that is not simply red in the individual DeltaO 13 and 18 records this should not really agree with model simulated variability?) it would be good to see a comparison of variability prior to confirming conclusions. Also, what analysis method was used, was existing code used following one of the recent papers (Ribes e.g.) or was a simple regression conducted? more should be said here and there definitely needs to be a comparison of simulated control and residual variability of the approach to ensure that the method is self consistent. also the scaling between proxy data (black in fig 4) and model data and the rationale of that comparison should be explained.

3) the spatial scale at which the detection and attribution analysis is conducted is not clear to me. figure 1 shows the location of the cave and a regional station measurement of temperature yet the variability is much larger than that of the wider CRUTEM region suggesting that there are strong regional effects. the regional water cycle and precipitation regionally can change for a lot of reasons, not all of them climatic. i am not fully convinced that the change seen in the station data and proxies is entirely climatic based on what i can see in the paper. Also, the history prior to the 1980s shows quite a different evolution of streamflow vs rainfall and P-PET which suggests that other influences may be at work (figure 1) - could that be discussed more clearly? generally, it is tricky to compare local precipitation related records to larger scale climate - can you rule out nonclimatic influences like land use change?

detail comments:

what is the ANA in l 129? (i think its explained in the supplement)

l 196 reading later in the ms i think the comparison of trends involves all variables in z scores. however the figure shows some in other units please clarify in text that the slopes are in z scores.

l 301 i think you have said pretty much the same about use of a D+A method a bit further up in the ms, this is a better place for it. maybe remove above. It needs to be clear what spatial representation of P-PET you use in your comparison.

The supplement gives the models but not the ensemble sizes. it would be good to mention these.

figure 4 caption: how is the detection and attribution analysis conducted - Hist vs Hist-nAT but then what is the role of GHG? and what is nat-GHG in the caption? are you using 3 fingerprints but then there is overlap between GHG and Hist? overall the detection and attribution approach could use a lot more clarity including size of region of model data used and scaling between P-PET and Onca2. (unless that is drawing on figure 3 which might be the case? could this be mentioned?)

fig. 5: why does the combination of the speleothems in the composite remove the strong variability in the individual ones - wouldn't the temperature anomaly be then better suited for attribution (although less exciting) due to much more homogeneous variability? unless the climate model P-PET variability is similar to that in figure 5...which i doubt

l 450: i am not convinced by analyzing P-PET from 1979 on and all else from 1900 this looks like cherrypicking to me.

some little typos e.g. detected in l 179, a few more too so maybe careful proofreading by a native speaker may be useful

RESPONSE TO REVIEWER COMMENTS

We are grateful to the reviewers for their thoughtful and encouraging comments on our manuscript NCOMMS-23-00133. We addressed the reviewer comments below in **bold blue text**.

Reviewer #1 (Remarks to the Author):

Strikis et al. present a new multiproxy record of rainfall and evaporation from stable isotope (d18O, d13C) and trace element (Mg/Ca, Sr/Ca, Ba/Ca) measurements from two speleothems located in central-eastern Brazil. Overall, the paper is quite well-written and structured, and the conclusions are well thought out. The record itself is also of high quality. The problem, however, is that many assumptions are made re: the controls on the speleothem proxies. Specifically, attempts by the authors to quantify P-PET using speleothem proxies that may have different temporal resolutions to observations and model simulations. In addition, efforts to calibrate (or assess 'linear congruence') the speleothem proxies using observations (via linear regression) can pose problems when factoring in the inherent seasonal biases associated with each proxy. For example, the d18O may be more sensitive to summer monsoon rainfall while the interannual trace element variability may be explained by dry season hydroclimate. These issues should be addressed in the manuscript.

Major corrections:

(1) The calibration of the geochemical proxies (which have not been shown to be recording annual/seasonal variability) against climate model output using linear regression. Aside from the fact that the different geochemical proxies have their own seasonal biases, the authors make little mention of the fact that the speleothem proxies may be (likely) recording multi-year variability given karst mixing. Hence, conducting a linear regression of this type from data of (likely) varying resolution is problematic and warrants further discussion in the manuscript.

Answer:

Indeed, the karst aquifer tends to buffer the seasonal and interannual variability by mixing the recharge solution leading to a multi-year variability. At Onça cave, where the thickness of the epikarst reservoir reaches more than 50 meters, this could lead to an important bias in the interannual variability. Therefore, in this study, we do not explore the interannual variability of rainfall proxies. Accordingly, we focus our discussion on the multi-year to decadal trends. Notwithstanding, seasonal variations of the cave environment related to temperature and relative humidity drive intra-annual variability as we demonstrate in Figure 1 below.

Figure 1 – Top: Comparison between seasonal Mg/Ca variation and cave relative humidity in experiment of calcite deposition at Onça Cave; Bottom: Comparison between Mg/Ca measurements on the Onça2 speleothem and potential evaporation measured at nearby local meteorological station (INMT 83386).

We have added a discussion of the results from this calcite deposition experiment in the manuscript (lines 189-193; 273-277; Method sections; 503-518) and included a Figure in the supplementary material (Fig. S8).

(2) The outdated equation used to calculate PET (using both observations and model output) without consideration of its uncertainties given that it relies heavily on temperature alone. It is well known that evaporative flux is also influenced by factors other than temperature, such as wind speed. Why did the authors only use this rather outdated equation to calculate PET and

not test other more updated equations? By doing so may have an influence on their findings. There is a rich body of work from the hydrology community looking at global and regional changes in PET from, for example, satellite data and using more updated equations. A deeper dive into this literature /research may benefit the study.

Answer:

This point raised by the reviewer is valid indeed. We opted to use the Thornthwaite equation because it offers the possibility to extrapolate the calculation of PET to periods prior to 1960 as this method uses only temperature. Hence this method allows reconstructing the PET using data from PHYDA or CMIP6.

We tested the PET obtained from the Thornthwaite equation using surface temperature data from INMET stations with the PET calculated using the Food and Agricultural Organization of the United Nations (FAO) Penman–Monteith method (formerly FAO-56 Penman–Monteith) (Allen et al., 1998; Raes, 2012) based on gridded data from ground-based stations provided by (Xavier et al., 2016, 2022). The data set from (Xavier et al., 2016, 2022) integrates data from ground-based station from the Instituto Nacional de Meteorologia (INMET) and the Agência Nacional de Águas (ANA).

In Figure 2 we compare the PET calculations using Thornthwaite (1948) presented in the original manuscript and the PET based on the Penman–Monteith method using gridded data from Xavier et al. (2022). To obtain values from PET from CMIP6 experiments we first calculated the PET using Thornthwaite (1948) from the simulated temperature of the different experiments and then we applied the monthly correction coefficient, as suggested in Aschonitis et al. (2022), to correct the PET from Thornthwaite (1948) (Figure 4 of the manuscript).

As suggested by the Reviewer #1, we ultimately decided to replace the PET values obtained from Thornthwaite (1948) with those calculated using the FAO-56 Penman–Monteith method. We corrected Figure 1 of the manuscript accordingly and adjusted the description of the methodology presented in lines 118-119 in the Section: “Study location and methodological approach”.

Figure 2 – Comparison between PET calculated using the Thornthwaite (1948) equation (black) and the FAO-56 Penman–Monteith (red) method based on gridded data from ground based-stations compiled from Xavier et al. (2022).

Questions/concerns re: the P-PET trends and methods used to calculate PET:

(1) The P-PET curve is remarkable similar to the P curve due to the very minor changes in PET. Hence, from the left panel it seems that most of the changes in water balance over recent decades is due to changes in precipitation. Is this correct? If so, the observed trends in speleothem geochemical proxies can be attributed to water deficits due to rainfall changes and not necessarily increased PET.

We thank the reviewer for bringing this issue to our attention. We recalculated the PET using gridded data from ground based stations (Xavier et al., 2020). As shown in Figure 3, evapotranspiration plays a significant role in the water deficit since 1979, as P-PET is characterized by a decreasing trend of $-12.5 \pm 2.7 \text{ mm.yr}^{-1}$, which is nearly 60 % more negative than the precipitation trend of $-7.1 \pm 1.9 \text{ mm.yr}^{-1}$ (both regression lines present p-values < 0.001 for a standard F-test). Conversely, the decreasing trend of precipitation is of course also important. We emphasize this aspect in the manuscript (lines 204-207):

“As shown in Fig. 1, the good correspondence between the negative trend observed in the streamflow of local rivers and the regional precipitation indicates that rainfall plays an important role in contributing to the hydrologic deficit. The persistently low precipitation since the 1990s in particular, has contributed to increased hydrologic stress in recent years”.

On the other hand, it is very important to note that proxies explored in this speleothem are particularly sensitive to water-atmosphere equilibration processes, as we observe in the Mg/Ca record. To better assess the role played by cave relative humidity in driving Mg/Ca variations, we added results from a monitoring experiment of calcite deposition performed at Lapa da Onça Cave to the supplementary material (Supplementary Figure S8).

Figure 3 - Comparison between annual regional precipitation (mm.yr⁻¹) (blue line) with the calculated hydrologic balance based on P-PET (mm.yr⁻¹) (red line). The potential evapotranspiration (PET) was calculated based on the FAO-56 Penman–Monteith equation.

We also emphasize the role played by evapotranspiration in the results section on lines 211-219: “The significance of the regression coefficients calculated for each variable are confirmed by an F-test, presenting p-values < 0.001. From 1979 to 2016 variations in streamflow of local rivers show a mean decrease of 22% per decade. During the same period, local precipitation presents a decreasing rate of only 7% per decade (70 mm.decade⁻¹) while P-PET presents a decreasing trend of 18 % (125 mm.decade⁻¹), indicating that changes in evapotranspiration are an important component of this drought. These comparisons suggest that the negative trend observed in the river discharge after the 1980s is partially a result of the increased evapotranspiration demand associated with the observed increase in local temperature.”

(2) As the authors are aware, there are numerous methods for calculating PET. Indeed, there is a body of literature that shows that temperature-based formulas such as the Thornthwaite (1948) equation are of low performance given that temperature alone cannot properly describe evaporative flux. As a result, studies have shown significant differences in PET estimates using Penman-Monteith-based versus temperature-based models. The authors should refer to a recent study by Aschonitis et al. (2022) (<https://essd.copernicus.org/articles/14/163/2022/essd-14-163-2022-discussion.html>) and references therein for a detailed analysis of the inherent

flaws in using the Thornthwaite equation in estimating PET and methods used to account for these flaws. In this regard, the authors should describe these uncertainties given that their conclusions primarily rest on assumptions drawn from their PET calculations. In addition, the authors should, in the very least, test the sensitivity of their results, including their model-proxy attribution analyses, using more than one method for calculating PET.

We agree with the reviewer that this is a relevant discussion and we thank the reviewer for pointing this out to us. Particularly, we believe that the best representation of the evaporative demand is recorded by Piche evaporimeter measurements, providing field measurements of the potential evaporative demand. However, we opted to use the Thornthwaite equation because it offers the possibility to extrapolate the calculation of PET to periods prior to the 1960s, given that this method uses only temperature. Hence this method allows us to reconstruct the PET using PHYDA or CMIP6 output. On the other hand, the PET calculated using the Thornthwaite 1948 equation can be underestimated, particularly in arid climates. Therefore, we tested the PET obtained based on the Thornthwaite equation using surface temperature data from INMET stations, with the PET calculated using the Food and Agricultural Organization of the United Nations (FAO) Penman–Monteith method (formerly FAO-56 Penman–Monte) (Allen et al., 1998; Raes, 2012) based on gridded data from ground-based stations provided by Xavier et al. (2016; 2020). The data set from Xavier et al., (2016; 2022) integrates data from ground-based stations from the *Instituto Nacional de Meteorologia* (INMET) and the *Agência Nacional de Águas* (ANA). Unlike the Thornthwaite (1948) method, the PET calculated based on the Penman–Monteith equation uses observational observed daily maximum and minimum temperature, daily solar radiation, relative humidity, and wind speed at 2m height.

We followed the method of Aschonitis et al. (2022) and calculated a monthly correction coefficient (Eq 1).

$$C_{th,i} = E_{r,i}/E_{p,i}.$$

Eq. 1

where $C_{th,i}$ is the correction coefficient for the monthly i ; $E_{r,i}$ is the potential evapotranspiration using the FAO-56 Penman-Monteith formula for the month i and $E_{p,i}$ is the potential evapotranspiration based on Thornthwaite equation for the month i . In Figure 4 we present the comparison between mean monthly values of PET calculated based on Thornthwaite (1948) (E_p) and the FAO-56 Penman-Monteith equation (E_r). The red curve refer to the monthly correction coefficient ($C_{th,i}$):

Figure 4 - Variation of mean monthly values of PET using Thornthwaite (1948) (green) and FAO-56 Penman-Monteith (orange) calculated for the area between 14-17° S and 43-45° W using gridded data derived from ground-based stations from Xavier et al. (2022). The red curve represents the ratio ($C_{th,i}$) between PET (FAO-56) and PET (Thornthwaite, 1948).

The resulting monthly $C_{th,i}$ (Eq. 1) was applied to the PET calculated using the Thornthwaite (1948) equation as demonstrated in Figure 5. The correction of the PET as suggested by the reviewer was incorporated by applying the correction factor to the calculations of PET and P-PET from temperature simulations from CMIP6 (Figure 4 of the manuscript).

Yet, following the suggestion of the Reviewer, we opted to use the PET obtained from the FAO-56 Penman-Monteith method when comparing streamflow and the hydrologic balance, as explored in Figures 1 and 3 of our manuscript.

Figure 5 – Left: Comparison between PET calculated using simple Thornthwaite (1948) equation (red), the PET calculated using the FAO-56 Penman-Monteith (black) and the corrected Thornthwaite (1948) values using the monthly $C_{th,i}$ according to Aschonities et al. (2022); Right: simple linear regression analysis between the mean monthly values of PET from Thornthwaite (1948) and FAO-56 Penman-Monteith using a linear least-squares regression method.

Minor corrections:

Line 70: “Drought events **are prone to** affect the chemistry...”

We accept the suggestion of the reviewer and changed the sentence to:

“Drought events can affect the chemistry...”

Line 74: “**Nevertheless,** few paleoclimate studies have...”

We accept the suggestion of the reviewer and changed the sentence to:

“However, few paleoclimate studies have ...”

Line 97: “...and trace elements **concentration** in speleothems from a cave have...”

We accept the suggestion of the reviewer and changed the sentence to:

“...and trace elements **concentrations** in speleothems from a cave ...”

Line 98: “Our results provide robust evidence that the current **aridification** trend **observed** has no precedent ...”

We accept the suggestion of the reviewer and changed the sentence to:

“Our results provide robust evidence that the current **drying** trend has no precedent ...”

Line 134: “hydroclimate variability obtained from speleothem record was compared ...”

We accept the suggestion of the reviewer and changed the sentence to:

“hydroclimate variability obtained from our speleothem record was compared ...”

Line 171: “The record from Onça Cave is composed of two speleothems, named Onça2 and Onça4 ...”

Reviewer comment: *“Can you confirm that it is annually-resolved? If so, how do you deconvolve the varying seasonal influences on the different proxies? For instance, I would expect the d_{18O} signal to be dominated by the summer monsoon rainfall, whereas the water balance proxies (i.e., trace elements) to have a dry season bias when PCP would be enhanced. This is important given that the authors use these data in an attribution analysis using climate model output. I assume they use annual”.*

Answer: The resolution of some of the proxies used in the Onça speleothems, like trace elements, permit the reconstruction of seasonal variability as we demonstrate in Figure 6 (not included in the paper). However, the growth rate is not suitable to extract such information from the oxygen isotope record. Nonetheless, we can state with certainty that the chronological control and the sampling resolution provide a reconstruction with an annually resolved chronology.

Figure 6 – High-resolution variation of the Mg/Ca record from the Onça2 stalagmite. The red lines indicate the annual chronology based on layer-counting.

However, the seasonal variability should be interpreted cautiously. The dripping rate tends to be a first-order driver of the physical-chemical processes that take place between the dripping solution and the cave atmosphere. For instance, the temperature effect on the $\delta^{18}\text{O}_{\text{calcite}}$ is largely dependent on the exchange time of water re-equilibration between the dripwater and the cave atmosphere (Casteel & Banner, 2015). The mechanism that drives trace element variability, the so-called “*prior calcite precipitation*”, is also affected by the dripping rate. On the other hand, because of the reservoir effect, cave drippings frequently present non-linear responses to the events of hydrologic recharge. Thus, the interannual variability of the proxy record may not reflect a straightforward function of the seasonal environmental variability. However, at multi-annual scales these effects tend to be minimized. That is why, in this paper, we explore trends observed on near-decadal timescales, rather than interannual variations, whose relative amplitude may be biased by the non-linear response of the karst hydrology. We hope to better address this question in future publications focused on Onça cave monitoring.

Lines 197-198: “Conversely, the calculated P-PET index shares nearly the same trend of the streamflow (-0.046 ± 0.005) (Fig 1b).”

Reviewer comment: “While there is no doubt that the trends are very much similar since 1980, why didn't the authors also compare the trends for the entire observational period? By not doing so gives the impression that the authors cherry-picked the year 1980 as this is where the trends look the most similar. To avoid criticism, the authors should compare trends for the whole period and explain reasons for any mismatch.”

Answer: Indeed, the selection of a starting point to calculate the regression may seem arbitrary based on how it was presented in the first version. In this study we use the detection change-point algorithm from Lavielle et al. (2005) and Killic er al. (2012) in streamflow time series to assess the timing of the onset of the regionally decreasing trend. By applying this method, we identified the year 1979 as the best reference for the onset of the drying trend. To clarify this aspect, we added a detailed explanation in the methods section and a Figure in the supplementary material.

See text added in Section “Study location and methodological approach” (lines 132-134) and the topic “Change point detection” in the “Method” section (lines 489-502) and new supplementary Figure S2.

Lines 204-205: “These comparisons suggest that the negative trend observed in the river discharge after 1980s is partially a result of the increased evapotranspiration demand ~~associated to the~~ **observed** with increasing local temperatures.”

Answer:

We accept the suggestion of the reviewer and changed the sentence to:

“These comparisons suggest that the negative trend observed in the river discharge after 1980s is partially a result of the increased evapotranspiration demand with increasing local temperatures”.

Lines 207-209: “Furthermore, a strong trend towards aridification over central-eastern South America after 2010 is also consistent with model projections, that point to a severe decrease in soil moisture, even under the moderate RCP 4.5 emission scenario, as a consequence of an increased evapotranspiration (Dai, 2012).”

Reviewer comment: Is there a more recent study to cite given the authors are talking about trends after 2010?

Answer: We accept the suggestion of the reviewer and we added a more recent publication that points to the role of increased evapotranspirative demand in the drying trend observed over central-eastern South America: we added Chagas et al. (2022) and Cuartas et al. (2022).

Line 247 – Caption Figure 2: “... and cave drip water from five monitoring sites (P1 to P5)”

We corrected this sentence according to the reviewer’s suggestion.

Line 256: “... processes like the ~~prior calcite precipitation~~ (PCP) on the control of geochemistry...”

Reviewer comment: You previously defined this acronym earlier in the manuscript. No need to define it again.

We corrected this sentence according to the reviewer’s suggestion.

Line 268-269: “... demand, which ultimately, ~~is~~ are driven by rising temperature ...”

We corrected this sentence according to the reviewer’s suggestion.

Line 270 “... we compare ~~the~~ regressed the z-scored time series ...”

We corrected this sentence according to the reviewer’s suggestion.

Reviewer #2 (Remarks to the Author):

This paper presents an analysis of speleothem records in central Brazil that the authors argue relate to changes in the watercycle in the region. The authors conduct a careful analysis of the role of precipitation and evaporation in the data and reconstruct a changing ability to meet evaporative demand from the data. A detection and attribution analysis then concludes that the driver of this change is anthropogenic. Overall it looks like there are some interesting findings here, but the approaches could be explained more clearly and I have some concerns about what exactly was done (which isnt everywhere clear either).

Thus I have the following comments

1) I like the careful analysis of the role of the various contributors to hydrological changes and the use of multiple proxies. I dont quite understand why the individual proxies ($\delta^{13}\text{C}$ and $\delta^{18}\text{O}$) have such strong variability while the combined anomaly which is interpreted as temperature anomaly (figure 5) has very robust and consistent variability. Can this be more clearly explained? and is the detection attribution analysis conducted on the components - it seems like this from figure 4 - which is interpreted as P-PET. What is the uncertainty in the calibration of one to the other - does this need to be considered? I doubt that the variability in the individual components is consistent with climate model variability.

Answer: The strong variability recorded in $\delta^{13}\text{C}$ and $\delta^{18}\text{O}$ observed in the Figure 5 (presented in the manuscript) may derive from two different aspects: i) the hydrogeochemical response of the Onça 4 speleothem to abrupt changes in cave water recharge or ii) past high-amplitude events that took place during the last millennium, such as large volcanic eruptions, for instance in 1452, 1600 and 1783 CE (Kuwae, Huaynaputina and Laki eruptions) that are not explored in this paper. The nearly 200 years of overlap between the Onça2 and Onça4 speleothem presented in the Supplementary Figure S12 suggests that both speleothems present a similar isotopic response at decadal to multi-decadal time-scales. We therefore argue that the isotopic excursions most likely reflect abrupt short-term wet events that have no counterpart with the wet events recorded during the XIX century.

Regarding the detection and attribution analysis, the comparison explored between $\delta^{18}\text{O}$ and the regional hydrologic balance takes advantage of the fact that regional rainfall amount is known to be a first-order driver of the oxygen isotope variability of the speleothems. Such a relationship with the rainfall amount can be observed in Figure 5 (presented in the manuscript), indicating that the isotope record is characterized by a higher variability than the trace elements (Mg/Ca).

2) The authors find a change in hydrological variables that is only reproduced if all forcing simulations are used to compare to the data. This is quite convincing as done, but the details are slim in the ms on the attribution approach - has a residual consistency test been conducted? (figure 5 suggests very strange variability that is not simply red in the individual $\delta^{13}\text{C}$ and $\delta^{18}\text{O}$ records this should not really agree with model simulated variability?) it would be good to see a comparison of variability prior to confirming conclusions. Also, what analysis method was used,

was existing code used following one of the recent papers (Ribes e.g.) or was a simple regression conducted? more should be said here and there definitely needs to be a comparison of simulated control and residual variability of the approach to ensure that the method is self consistent. Also the scaling between proxy data (black in fig 4) and model data and the rationale of that comparison should be explained.

Answer:

We acknowledge the valuable feedback provided by the reviewers regarding the statistical analysis of the residuals obtained from the Detection and Attribution (D&A) using simple least square regression. In response to this feedback, we have included the results of the standard F-test of residual consistency in this revised version.

To address this concern, we have made the following additions and modifications to the manuscript:

1. Method Section: We have included a description of the residual consistency analysis in the Method section. Specifically, the details of the analysis can be found in lines 485-488 of the Method section.

2. Supplementary Material: We have added a new table, Table S4, in the supplementary material. This table presents the p-values obtained from the F-test of the residual analysis between the simulated control and the speleothem proxies, specifically Mg/Ca and $\delta^{18}\text{O}$.

In general the residual of the regressions presents a normal distribution as we demonstrate in the figure

In the Climate Implications section, lines 346-349, we have also provided a rationale for the comparison between the simulated control and the speleothem proxies. We explain that this comparison is conducted to investigate the primary drivers of changes in the regional hydrologic balance and to assess the contribution of anthropogenic factors. Furthermore, we mention that the method employed is based on a least square linear regression between the geochemical proxies Mg/Ca and $\delta^{18}\text{O}$ of the speleothem and PET and P-PET simulated by an ensemble of CMIP6 models from the Detection and Attribution Model Intercomparison Project (DAMIP).

We believe that these additions and modifications address the reviewers' concerns and enhance the statistical analysis of the residuals obtained from the D&A using simple least square regression.

Figure 7 – Example of F-tests performed by the regression between z-scored values of Mg/Ca and model simulations from CMIP-6 used in the experiments nat-GHG (natural and anthropogenic forcings); nat-only (natural forcings only) and GHG-only (anthropogenic only forcing). The p-values obtained from the F-test of each regression are presented in the Table S4.

“To assess the linear congruence between annual mean streamflow flow with local rainfall and P-PET we used the period between 1979 to 2016. For the regression analysis between the speleothem geochemical proxies and CMIP6 simulations we used the period between 1900 and 2015. The resulting regression coefficients provide a scaling factor that is used to compare the

trends between time-series. The statistical significance of each regression were analyzed using the standard F-test (Allen & Tett, 1999). The proxies, i.e. $\delta^{18}\text{O}$ or Mg/Ca were used as the response variable (x) and the temperature simulations from the different model were used as the explanatory variable (y). The p-values of the F-test are presented in the supplementary Table S4.”

We wish to emphasize that the main goal of the D&A analysis is to identify the main drivers of the decadal drying trend, as captured in the trace element and isotope records. Regarding the method used for the D&A analysis, as we describe in the “Detection and Attribution Analysis” section, the calculations involved in this test are based on ordinary linear regressions: “A linear regression coefficient using the least-squares fit is then calculated, comparing proxy data against the forcings using a 5-yr running mean time-series”. The coefficient of the least-squares fit was calculated using OriginLab software.

3) The spatial scale at which the detection and attribution analysis is conducted is not clear to me. figure 1 shows the location of the cave and a regional station measurement of temperature yet the variability is much larger than that of the wider CRUTEM region suggesting that there are strong regional effects. the regional water cycle and precipitation regionally can change for a lot of reasons, not all of them climatic. I am not fully convinced that the change seen in the station data and proxies is entirely climatic based on what i can see in the paper. Also, the history prior to the 1980s shows quite a different evolution of streamflow vs rainfall and P-PET which suggests that other influences may be at work (figure 1) - could that be discussed more clearly? generally, it is tricky to compare local precipitation related records to larger scale climate - can you rule out nonclimatic influences like land use change?

Answer: We applied the detection and attribution analysis over the region we used to assess the hydrological balance, covering a 4° x 4° area (13° – 17° S; 42.5° – 46.5° W), centered over the cave site. We added a sentence on lines 328-329 to clarify this point.

Regarding the regional temperature, CRUTEM may actually underestimate the regional temperature. The data presented in the paper are based on simple arithmetic means of monthly mean temperature. The regionally averaged temperature record does not show significant differences when compared with gridded data from ground-based stations (Xavier et al., 2020). In Figure 1c we present a comparison between the gridded, ground-based record from Xavier et al. (2020) and the CRU TS 4.06 (Harrison et al., 2020), showing that the temperature variability is consistent among the records.

The cave lies in a protected area, a National Park, whose vegetation has been protected throughout the 20th century, partly due to the karst relief itself rendering farming difficult, if not impossible. We can therefore exclude human disturbance affecting the geochemical proxies. In order to clarify this aspect, we now briefly discuss this point on lines 165-169.

Finally, it is important to note that, the streamflow coherence among the stations is high (see Figure 8 below), even though the number of fluviometric stations is significantly smaller (just 4 stations) prior to the 1970s, in comparison to the time period from 1980 to 2016 (between 11 and 13 stations). However, as shown in Figures 3, 4 and 5 of the manuscript, the evaporative

forcing associated with temperature serves as the major driver of isotopic variability. This is particularly clear in Figure 5 of the manuscript.

Figure 8 - Time series of the streamflow of local rivers. The data is presented in z-score units for easier comparison among the rivers.

Nevertheless, similar trends of reduced streamflow may be observed when comparing rivers from protected areas, where land use tends to be minimal with those that cut across agricultural zones, where water extraction may be critical. For instance, in Figure 9 we compare the streamflow of two local rivers with different watersheds in terms of land use: Pandeiros and Coxá River. Pandeiros River lies in a protected area while Coxá River flows through an area with intense occupation, with farms presenting center-pivot irrigation at just a few meters distance from the river. On the other hand, we acknowledge that assessing the effect of human occupation would demand a careful and detailed analysis, which is beyond the scope of this study. We acknowledge that we cannot entirely rule out the effect of land use in affecting the steep reduction of the regional streamflow, particularly from the year 2000 onward, as pointed out in recent studies (Chagas et al., 2000). We added a brief discussion addressing these questions in lines 224-227.

Figure 9 - Comparison between streamflow of a river in a protected watershed (Pandeiros River in blue) with a river that flows through an agricultural zone with intense water extraction (Coxá River in red). The bold black line shows the mean regional streamflow.

detail comments:

what is the ANA in l 129? (i think its explained in the supplement)

Answer: ANA refers to Agência Nacional de Águas. This acronym is defined in line 117 and in the caption of Figure 1.

l 196 reading later in the ms i think the comparison of trends involves all variables in z scores. however the figure shows some in other units please clarify in text that the slopes are in z scores.

Answer: we corrected that adding a reference of the values in z-score (line 209)

l 301 I think you have said pretty much the same about use of a D+A method a bit further up in the ms, this is a better place for it. maybe remove above. It needs to be clear what spatial representation of P-PET you use in your comparison. The supplement gives the models but not the ensemble sizes. it would be good to mention these.

Answer: For the CMIP 6 we used the same area explored in the comparisons between proxies and the regional hydrologic balance using a square of 4x4 centered in the study area. We a lines 328-329: "The test was performed using data from natural and anthropogenically forced historical simulations (1850-2014) covering a region of 4 x 4° (13 – 17° S; 42.5 – 46.5° W) centered over the cave site, the same region used when calculating the hydrologic balance using instrumental records".

figure 4 caption: how is the detection and attribution analysis conducted - Hist vs Hist-nAT but then what is the role of GHG? and what is nat-GHG in the caption? are you using 3 fingerprints but then there is overlap between GHG and Hist? overall the detection and attribution approach could use a lot more clarity including size of region of model data used and scaling between P-PET and Onca2. (unless that is drawing on figure 3 which might be the case? could this be mentioned?)

Answer:

We thanks to the reviewer for raising these issues. In fact, there is some confusion regarding the acronyms of the experiments presented in the Figure captions. Overall we considered three forcings:

- **nat-GHG: natural and anthropogenic forcings, that include volcanic, solar and anthropogenic greenhouse gas forcing;**
- **nat-only: natural forcings only (solar irradiance and stratospheric aerosol forcing from volcanic eruptions)**
- **GHG-only: anthropogenic only forcing. Well-mixed greenhouse-gas-only historical simulations.**

We corrected the acronyms in the elements of Figures 4a and 4c.

Regarding the area investigated, we added a sentence in lines 358-329: “... covering a region of $4 \times 4^\circ$ ($13 - 17^\circ S$; $42.5 - 46.5^\circ W$) centered over the cave site, the same region used in calculation of the hydrologic balance using instrumental records”.

Regarding the scaling, in Figure 3 (presented in the manuscript) we compare the scaling between P-PET and the Onça2 proxies for the instrumental records, which allowed us to identify the evaporative forcings (temperature, evapotranspiration potential and evaporation potential) as the main drivers of trace element and isotope variability at decadal timescales.

fig. 5: why does the combination of the speleothems in the composite remove the strong variability in the individual ones - wouldn't the temperature anomaly be then better suited for attribution (although less exciting) due to much more homogeneous variability? unless the climate model P-PET variability is similar to that in figure 5...which i doubt | 450: i am not convinced by analyzing P-PET from 1979 on and all else from 1900 this looks like cherry-picking to me.

The top panel of Figure 5 (presented in the manuscript) shows a comparison between the full Mg/Ca record from the Onça2 speleothem extending back to 1735 CE with the temperature reconstructions. Thus, there is no composite shown in the top panel of Figure 5 (presented in the manuscript). In the lower panel, however, we do indeed, present a composite combining the Onça2 and Onça4 isotope records, which allows us to extend our reconstruction back to the 1283 CE. However, we do not observe a loss of the variability in the composite. In contrast, the composite presented in the lower panel reveals a period marked by the recurrence of abrupt events over the last millennium. On lines 379-383 we provide a brief description of those abrupt short-lived events whose meaning we opted to not explore further in this

publication as they are likely related to external forcing, such as large volcanic eruptions or solar irradiance, which is beyond the scope of this study.

Finally, reviewer #1 raised a similar question regarding the selected starting point in 1979. We therefore repeat our response to R1 below:

Indeed, the selection of a starting point to calculate the regression may seem arbitrary based on how it was presented in the first version. In this study we use the detection change-point algorithm from Lavielle et al. (2005) and Killic er al. (2012) in streamflow time series to assess the timing of the onset of the regionally decreasing trend. By applying this method, we identified the year 1979 as the best reference for the onset of the drying trend. To clarify this aspect, we added a detailed explanation in the methods section and a Figure in the supplementary material.

See text added in Section “Study location and methodological approach” (lines 132-134 in manuscript) and the topic “Change point detection” in the “Method” section (lines 491-504) and new supplementary Figure S2.

some little typos e.g. detected in l 179, a few more too so maybe careful proofreading by a native speaker may be useful.

References

- Allen, M. R., & Tett, S. F. B. (1999). Checking for model consistency in optimal fingerprinting. *Climate Dynamics*, 15(1–2), 419–434. <https://doi.org/10.1007/s003820050291>
- Aschonitis, V., Touloumidis, D., Ten Veldhuis, M. C., & Coenders-Gerrits, M. (2022). Correcting Thornthwaite potential evapotranspiration using a global grid of local coefficients to support temperature-based estimations of reference evapotranspiration and aridity indices. *Earth System Science Data*, 14(1), 163–177. <https://doi.org/10.5194/essd-14-163-2022>
- Casteel, R. C., & Banner, J. L. (2015). Temperature-driven seasonal calcite growth and drip water trace element variations in a well-ventilated Texas cave: Implications for speleothem paleoclimate studies. *Chemical Geology*, 392, 43–58. <https://doi.org/10.1016/j.chemgeo.2014.11.002>
- Raes. 2012. The ETo Calculator. Food and Agriculture Organization of the United Nations, FAO: Rome.
- Xavier, A. C., King, C. W., & Scanlon, B. R. (2016). Daily gridded meteorological variables in Brazil (1980–2013). *International Journal of Climatology*, 36(6), 2644–2659. <https://doi.org/10.1002/joc.4518>
- Xavier, A. C., Scanlon, B. R., King, C. W., & Alves, A. I. (2022). New improved Brazilian daily weather gridded data (1961–2020). *International Journal of Climatology*, 42(16), 8390–8404. <https://doi.org/10.1002/joc.7731>

REVIEWER COMMENTS

Reviewer #1 (Remarks to the Author):

I am satisfied with the revisions and commend the authors for their efforts in addressing my comments and suggestions. Thus, I can now recommend the paper for publication.

Reviewer #2 (Remarks to the Author):

Thank you for considering my comments carefully and i find the revised ms much clearer (and thanks for the z-scores in some literature (non hydro) this is call standardized but as long as its explained it is a good choice.

There are some small issues remaining that i would like to see addressed:

1) the significance testing is still not quite clear to me. You relate to Allen + Tett's paper that is based on an F-test, but it will only work on pre-whitened data (i.e. method with optimizing - it is not mentioned that this was done?). So the method text still doesnt quite give enough information, and sounds more like a generic F test is conducted which would not be ok as the method will need to detect the role of forcing against decadal and multidecadal variability. Any method used needs to take that into account. most approaches just use samples from either model control simulations, or the within-ensemble variability in the forced runs providing the fingerprints (e.g. all GHG+nat runs individually minus the ensemble mean, which gives a sample of variability, corrected by a factor for the variance loss from subtracting the ensemble mean which is still noisy (I think its $\sqrt{n/(n-1)}$). please clarify the uncertainty estimate in the methods section - there is not much there and the rebuttal reference gives the wrong line numbers. Alternatively, maybe you used available software which would be helpful (and fair) to say, please clarify this. I 475 mentions just a standard F test. the analyses you made to support robust residuals is quite nice and could go into a supplement to support reliable results.

2) thanks for clarifying the forcing simulations. i am not quite clear if the anthropogenically forced simulations include aerosols and other anthropogenic forcings - that is usually the convention. can you clarify in text and fix? normally, HIST includes all forcings including other anthropogenic than ghg, such as aerosols, ozone, land use... while nat include solar and volcanic. this could use a quick check. (e.g, I 325)

3) this is a new question sorry - i think its addressed but not clear from text: you illustrate nicely how all your timeseries show similar trends in z scores. However, this could happen for a lot of reasons, while in your case they also covary strongly as obvious from the figure (but not quantified at least not in caption) - i recommend to highlight that too. that way you have many degrees of freedom supporting your proxy.

Minor comments:

- Abstract, l 95 and on: would prefer not to call detection and attribution 'formal detection and attribution' as i dont think thats a useful distinction and it may also get confused with approaches using some of the available software and fairly sophisticated methods while here i think a multi regression only is used (which is fine for timeseries particularly and more transparent!). If you have used available software and a prewhitening/optimal fingerprint approach please say so.

l 74 has a typo or strange format in the reference 20

i suggest you refer to this paper in the introduction which draws related conclusions but for different and larger regions: Marvel, K., B.I. Cook, C.J.W. Bonfils, P.J. Durack, J.E. Smerdon, and A.P. Williams, 2019: Twentieth-century hydroclimate changes consistent with human influence. *Nature*, 569, no. 7754, 59-65, doi:10.1038/s41586-019-1149-8.

RESPONSE TO REVIEWER COMMENTS

We are grateful to the reviewers for their thoughtful and encouraging comments on our manuscript NCOMMS-23-00133. We addressed the reviewer comments below in **bold blue text**.

Reviewer #1 (Remarks to the Author):

I am satisfied with the revisions and commend the authors for their efforts in addressing my comments and suggestions. Thus, I can now recommend the paper for publication.

Authors comments: We thank the reviewer for the supportive comments.

Reviewer #2 (Remarks to the Author):

Thank you for considering my comments carefully and I find the revised ms much clearer (and thanks for the z-scores in some literature (non-hydro) this is call standardized but as long as it's explained it is a good choice.

There are some small issues remaining that I would like to see addressed:

- 1) the significance testing is still not quite clear to me. You relate to Allen + Tett's paper that is based on an F-test, but it will only work on pre-whitened data (i.e. method with optimizing - it is not mentioned that this was done?). So, the method text still doesn't quite give enough information, and sounds more like a generic F test is conducted which would not be ok as the method will need to detect the role of forcing against decadal and multidecadal variability. Any method used needs to take that into account. Most approaches just use samples from either model control simulations, or the within-ensemble variability in the forced runs providing the fingerprints (e.g. all GHG+nat runs individually minus the ensemble mean, which gives a sample of variability, corrected by a factor for the variance loss from subtracting the ensemble mean which is still noisy (I think its $\sqrt{n/(n-1)}$). Please clarify the uncertainty estimate in the methods section - there is not much there and the rebuttal reference gives the wrong line numbers. Alternatively, maybe you used available software which would be helpful (and fair) to say, please clarify this. I 475 mentions just a standard F test. the analyses you made to support robust residuals is quite nice and could go into a supplement to support reliable results.

Answer:

Dear Reviewer #2

We thank you for your insightful comments on our approach to the D&A test, specifically regarding the need to account for the within-ensemble variability. To address the importance

of multidecadal variability in our analysis and the problems related to the potential autocorrelation we tested two different approaches that we present below.

In the first approach we explore the problem of autocorrelation by pre-whitening the time series, ensuring that residuals from our regression models don't show significant autocorrelation, making our subsequent beta confidence intervals more robust.

We applied a 5-year running average to our datasets, enabling us to mitigate the influence of shorter-term variability and highlight the underlying multidecadal trends. Additionally, based on the Reviewer's suggestion, we incorporated the within-ensemble variability from our model experiments into our analysis. To do so, we regressed the z-score data of our proxies (e.g., Mg/Ca and $\delta^{18}\text{O}$) against individual runs of model experiments (NAT, GHG+NAT and NAT), and factored in the within-ensemble variability by weighting it appropriately in our beta confidence interval.

However, the effect of pre-whitening the data on the trend of the speleothem proxies can be assessed in the comparisons presented in **Figure 1** (this document). As we demonstrate in the **Figures 1a** and **1c**, the effect of pre-whitening is particularly significant in the case of $\delta^{18}\text{O}$.

Figure 1 – Comparison of Beta coefficient and the significance of the regression (F-test) with (red) and without (black) pre-whitening: a) Mg/Ca from Onça2 speleothem; b) ensemble mean times series of PET from GHG+NAT CMIP6 experiment; c) $\delta^{18}\text{O}$ from Onça2 speleothem; d) ensemble mean times series of P-PET from GHG+NAT CMIP6 experiment.

As the pre-whitening method proposed by the reviewer has an outsized effect on the slope of the proxy time series, we tested an alternative approach to the detection & attribution analysis. This may be because GHG forcing, while steadily increasing through the 19th and 20th centuries, can modulate internal multidecadal climate variability. By pre-whitening the data, we may thus be removing an important part of the anthropogenic signal.

Detection and Attribution analysis generally considers the intra-ensemble variability and makes use of truncated spatial EOFs to filter the climate signal. Thus, to overcome the problems related to autocorrelation or dependency among observations (non-independent and identically distributed data) and retain the multidecadal signal, e.g. the long-term trend, we tested a different approach using the minimax regression algorithm described in Xu et al. (2019) using 20 Monte-Carlo simulations.

The regression lines for Mg/Ca and $\delta^{18}\text{O}$ with their corresponding scaling factors (β) are shown in **Figures 3 and 4 (this document), respectively. While the GHG and GHG+NAT scaling factors are nearly identical, the NAT forcing scenario results in near-zero scaling factors, indicating a very limited and insignificant influence of natural forcing, once following our hypothesis, the trend is modulated by due to GHG forcing, as seen in the Mg/Ca and $\delta^{18}\text{O}$ time series.**

Figure 3 – Regression lines with their corresponding scaling factors (β -coefficient) from Mg/Ca vs. PET ensemble medians for NAT, NAT+GHG and GHG experiment. The whisker plot shows the β -coefficient obtained from 20 Monte Carlo simulations estimated based on the regression between proxy and P-PET from each ensemble member.

Figure 4 – Regression lines with their corresponding scaling factors (β -coefficient) from $\delta^{18}\text{O}$ vs P-PET ensemble medians for NAT, NAT+GHG and GHG experiment. The whisker plot shows the β -coefficient obtained from 20 Monte Carlo simulations estimated based on the regression between proxy and P-PET from each ensemble member.

Next, in **Figure 5** (this document) we provide a scaling factor comparison. The β -coefficients from **Figures 3** and **4** are replotted as whisker plots in the way they are presented in the paper. Both GHG and GHG+NAT reach scaling factors close to 1, indicating the strong dependence of the trend in the proxy time series on GHG and GHG+NAT forcing. The NAT scenario on the other hand in both instances (Mg/Ca & $\delta^{18}\text{O}$) has a low scaling factor close to zero indicating that the trend observed in the proxy time series is inconsistent with the NAT scenario.

Figure 5 – Box plots presenting the scaling factor. Left: Scaling factors of PET (based on individual runs from CMIP6 experiments) vs. speleothem Mg/Ca. Right: same as left panel but for P-PET vs. $\delta^{18}\text{O}$.

To test whether the scaling factors $\beta \sim N(\hat{\beta}, \hat{\sigma}^2)$, are statistically significantly different between GHG+NAT, GHG and NAT scenarios, we apply a t-test to compare two normal populations, assuming unequal and unknown variance among the coefficients or scaling factors. So, the t-test is given by:

$$T = \frac{\overline{\beta_1} - \overline{\beta_2}}{\sqrt{\frac{S_1}{n_1} + \frac{S_2}{n_2}}}$$

And the degrees of freedom of the t-distribution is given by:

$$\nu = \frac{(A^2 - B^2)}{\frac{A^2}{(n_1 - 1)} + \frac{B^2}{(n_2 - 1)}}$$

$$A = \frac{S_1^2}{n_1} \quad B = \frac{S_2^2}{n_2}$$

Mg/Ca				$\delta^{18}\text{O}$		
Hypotheses	T	v	p-value	Hypotheses	T	v
$H_0: \beta_{\text{GHG_only}} = \beta_{\text{nat_GHG}}$	-0.27	10	0.794	$H_0: \beta_{\text{GHG_only}} = \beta_{\text{nat_GHG}}$	-0.44	15
$H_0: \beta_{\text{GHG_only}} = \beta_{\text{nat_only}}$	5.77	13	0.0001	$H_0: \beta_{\text{GHG_only}} = \beta_{\text{nat_only}}$	4.57	12
$H_0: \beta_{\text{nat_GHG}} = \beta_{\text{nat_only}}$	10.17	12	>0.0001	$H_0: \beta_{\text{nat_GHG}} = \beta_{\text{nat_only}}$	4.45	11

Hypotheses	T	v	p-value	Hypotheses	T	v
$H_0: \beta_{\text{GHG_only}} = 1$	3.83	8	0.005	$H_0: \beta_{\text{GHG_only}} = 1$	9.33	8
$H_0: \beta_{\text{nat_GHG}} = 1$	11.17	8	>0.0001	$H_0: \beta_{\text{nat_GHG}} = 1$	7.19	8
$H_0: \beta_{\text{nat_only}} = 0$	0.49	8	0.6384	$H_0: \beta_{\text{nat_only}} = 0$	-0.56	8

The p-value based on t-test statistics rejects the null hypothesis at the level of 0.05, indicating that the scaling factor for the NAT simulation is different from GHG and GHG+NAT scenario. The GHG and GHG+NAT scenarios appear statistically indistinguishable, highlighting the significant role of forcing in influencing the observed trend of proxies. This is further supported by hypothesis testing, where the scaling factors being equal to 1 favor the null hypothesis.

We made changes in the Method Section lines 485 to 493 (track-changes file), lines 500-507 and Results and Discussion Section lines 325-329 (track-changes file).

We added new figures (Figure S13 and S14) in the supplementary material depicting the output of the minmax algorithm using 20 Monte Carlo simulations for the calculation of the linear regression between the speleothem proxies Mg/Ca and $\delta^{18}\text{O}$ against the CMIP6 ensembles of PET and P-PET for GHG, GHG+NAT and NAT experiments. The p-values of the D&A analysis between proxies and CMIP simulations were added in supplementary material to Table S4.

References

- Allen, M. R. & Tett, S. F. B. Checking for model consistency in optimal fingerprinting. *Clim. Dyn.* 15, 419–434 (1999).
- Xu, Q. & Xuan, X. M. Nonlinear regression without i.i.d. assumption. *Probab. Uncertain. Quant. Risk* 4, (2019).

- 2) thanks for clarifying the forcing simulations. I am not quite clear if the anthropogenically forced simulations include aerosols and other anthropogenic forcings - that is usually the convention. can you clarify in text and fix? normally, HIST includes all forcings including other anthropogenic than ghg, such as aerosols, ozone, land use... while nat include solar and volcanic. this could use a quick check. (e.g, l 325)

Answer:

We thank the reviewer for pointing this out. We agree that this aspect was not clear in the text.

The experiments used in the paper comprise Tier 1 of the DAMIP which includes:

- Historical: *“These simulations are forced, based on observations, by evolving, externally imposed forcings such as solar variability, volcanic aerosols, and changes in atmospheric composition (GHGs and aerosols) caused by human activities.”* (Eyring et al. 2016).

- Hist-nat: *“These historical natural-only simulations resemble the historical simulations but are forced with only solar and volcanic forcings from the historical simulations, similarly to the CMIP5 historicalNat experiment.”* (Gillett et al. 2016).

- Hist-GHG: *“These historical greenhouse-gas-only simulations resemble the historical simulations but instead are forced by well-mixed greenhouse gas changes only, similarly to the CMIP5 historical-GHG experiment.”* (Gillett et al. 2016).

In order to better clarify these aspects, we made some minor changes in the text (**lines 332 to 338**; track change file):

“For this purpose, we selected three different experiments: i) Historical experiment (GHG+NAT) that includes externally imposed, naturally evolving forcings represented by volcanic aerosols, solar variability and anthropogenic greenhouse gas and aerosol forcings; ii) Hist-nat experiment (NAT) – this simulation is forced only with solar irradiance and stratospheric aerosol forcing from volcanic eruptions, and iii) Hist-GHG experiment (GHG) – forced with the well-mixed greenhouse-gas changes only.

- 3) This is a new question sorry – I think its addressed but not clear from text: you illustrate nicely how all your timeseries show similar trends in z scores. However, this could happen for a lot of reasons, while in your case they also covary strongly as obvious from the figure (but not quantified at least not in caption) – I recommend to highlight that too. That way you have many degrees of freedom supporting your proxy.

We thank the reviewer for pointing this out. We changed the caption in **Figure 3** (manuscript file) showing the values of the slope (β) of the z-score data. We did some minor changes in the text in the Result and Discussion Section, lines **307-309** and **312-313** (track-changes file). We also highlight that all the regressions performed in this test are statistically significant at $p < 0.001$ ($n = 71$) lines **301-303** (Figure captions) (track-changes file).

Minor comments:

- Abstract, l 95 and on: would prefer not to call detection and attribution 'formal detection and attribution' as i dont think thats a useful distinction and it may also get confused with approaches using some of the available software and fairly sophisticated methods while here i think a multi regression only is used (which is fine for timeseries particularly and more transparent!). If you have used available software and a prewhitening/optimal fingerprint approach please say so.

Answer:

We accept the suggestion of the reviewer and changed the sentences by suppressing the expression “*formal*” in the lines 36, 95, 315 of the manuscript.

- line 74 has a typo or strange format in the reference 20 I suggest you refer to this paper in the introduction which draws related conclusions but for different and larger regions: Marvel, K., B.I. Cook, C.J.W. Bonfils, P.J. Durack, J.E. Smerdon, and A.P. Williams, 2019: Twentieth-century hydroclimate changes consistent with human influence. *Nature*, 569, no. 7754, 59-65, doi:10.1038/s41586-019-1149-8.

Answer:

We corrected the format of the reference and added the suggested paper to the references section and now cite it in the text in lines 58 and 78 in the following sentences:

“The increased aridity observed in the region during the last decades has raised questions regarding the role of human-induced warming in driving changes in the hydrologic balance and how drought risk might be exacerbated by anthropogenic greenhouse gas emissions in the coming decades^{2,3,9,11}.”

“Reconstructing past climatic conditions in the region, prior to the period covered by instrumental records, is vital to document the full amplitude of natural climate variability and to quantify the contributions from external and internal forcings to regional hydrologic variability¹¹.”

Reference

¹¹ Marvel, K. et al. Twentieth-century hydroclimate changes consistent with human influence. *Nature* 569, 59–65 (2019)

Additional minor corrections

- We change the author order of the co-author Ernesto Tejedor and we correct his scientific affiliation
- We updated the cave map in the Supplementary Figure S4.
- We changed the title to “Modern anthropogenic drought in Central Brazil unprecedented during last 700 years” as the old title erroneously implied that the observed drought persisted throughout the entire Last Millennium, when in reality it emerged in response to anthropogenic forcing over the last 2 centuries.
- We added the section “Code availability” lines: **563-564** (track changes file)
- We added the section “Author contributions” lines: **736-741** (track changes file)
- We added Inclusion & ethics statement lines: **743-745** (track changes file)

REVIEWERS' COMMENTS

Reviewer #2 (Remarks to the Author):

The authors have addressed most of my comments. the pre-whitening was a misunderstanding - the way the text was phrased an expectation of a much more sophisticated analysis than has been done arose, and i gave the prewhitening as an example (in the 'formal' detection and attribution approaches the NOISE is usually prewhitened not the signal the latter would indeed not work!) the attribution is still somewhat opaque, i THINK you conduct a multi regression based on z score data (which is good) and you assume white noise in it [if not please clarify in final version, if yes consider explicitly saying so]. you may want to flag as an uncertainty in your analysis that climate variability is red but precipitation tends to be fairly white so your approach should be fine. i have no further comments, its a nice ms